# Live cell-lineage tracing and machine learning reveal patterns of organ regeneration

Oriol Viader-Llargués[1,2], Valerio Lupperger[3], Laura Pola-Morell[1,2], Carsten Marr[3]*, Hernán López-Schier[1,2]*

[1]Unit Sensory Biology & Organogenesis, Helmholtz Zentrum München, Neuherberg, Germany; [2]Laboratory of Sensory Cell Biology, Centre for Genomic Regulation, Barcelona, Spain; [3]Institute of Computational Biology, Helmholtz Zentrum München, Neuherberg, Germany

**Abstract** Despite the intrinsically stochastic nature of damage, sensory organs recapitulate normal architecture during repair to maintain function. Here we present a quantitative approach that combines live cell-lineage tracing and multifactorial classification by machine learning to reveal how cell identity and localization are coordinated during organ regeneration. We use the superficial neuromasts in larval zebrafish, which contain three cell classes organized in radial symmetry and a single planar-polarity axis. Visualization of cell-fate transitions at high temporal resolution shows that neuromasts regenerate isotropically to recover geometric order, proportions and polarity with exceptional accuracy. We identify mediolateral position within the growing tissue as the best predictor of cell-fate acquisition. We propose a self-regulatory mechanism that guides the regenerative process to identical outcome with minimal extrinsic information. The integrated approach that we have developed is simple and broadly applicable, and should help define predictive signatures of cellular behavior during the construction of complex tissues.
DOI: https://doi.org/10.7554/eLife.30823.001

*For correspondence:
carsten.marr@helmholtz-muenchen.de (CM);
hernan.lopez-schier@helmholtz-muenchen.de (HL-S)

**Competing interests:** The authors declare that no competing interests exist.

## Introduction

Understanding organogenesis, organ morphostasis and regeneration is crucial to many areas of biology and medicine, including controlled organ engineering for clinical applications (*Lancaster et al., 2013*; *Boj et al., 2015*; *Sato and Clevers, 2015*; *Willyard, 2015*). External tissues sustain continuous injury and must recurrently repair to maintain physiological function during the life of the organism (*Levin, 2009*). Structural reproducibility depends on the re-establishment of cell identity, number, localization and polarization. Two aspects of organ regeneration are the current focus of intense attention. First, how multiple cells interact to recapitulate organ architecture. Second, what is the mechanism that controls the correct reproduction of cell number and localization. Here we use the neuromasts of the superficial lateral line in larval zebrafish to gain a global perspective on sensory-organ regeneration. The neuromasts are ideally suited for this purpose because they are small and external, facilitating physical access and three-dimensional high-resolution videomicroscopy of every cell during extended periods. We have combined live single-cell tracking, cell-lineage tracing, pharmacological and microsurgical manipulations, and multidimensional data analysis by machine learning to identify features that predict cell-fate decisions during neuromast repair. Our comprehensive approach is simple and model independent, which should facilitate its application to other organs or experimental systems that are accessible to videomicroscopy. It should help reveal the basic rules that underlie how complex structures emerge from the collective behavior of cells.

## Results

### Complete neuromast ablation is irreversible in larval zebrafish

The neuromasts of the superficial lateral line in zebrafish are formed by a circular cuboidal epithelium of 60–70 cells (*López-Schier and Hudspeth, 2006*; *Ghysen and Dambly-Chaudière, 2007*; *Norden, 2017*). Mechanoreceptive hair cells occupy the center of the organ, whereas non-sensory sustentacular supporting cells are found around and between the hair cells (*Figure 1A*). A second class of supporting cell called mantle cells forms the outer rim of the organ. The invariant spatial distribution of these three cell classes generates a radial symmetry (*Figure 1B*) (*Pinto-Teixeira et al., 2015*). Neuromasts also have an axis of planar polarity defined by the orientation of the hair-cells' apical hair bundle (*Figure 1C*) (*Ghysen and Dambly-Chaudière, 2007*; *Wibowo et al., 2011*). In addition to this geometric organization, cell-class number and proportions are largely constant, with around 40 sustentacular, 8–10 mantle, and 14–16 hair cells. Non-sensory cells can proliferate, whereas the sensory hair cells are postmitotic (*López-Schier and Hudspeth, 2006*; *Ma et al., 2008*; *Cruz et al., 2015*; *Pinto-Teixeira et al., 2015*). Finally, a string of interneuromast cells connects each neuromast along the entire lateral-line system (*Figure 1A*) (*Ghysen and Dambly-Chaudière, 2007*). Previous studies have extensively characterized the regeneration of the mechanosensory hair cells (*Williams and Holder, 2000*; *Harris et al., 2003*; *López-Schier and Hudspeth, 2006*; *Hernández et al., 2006*; *Ma et al., 2008*; *Behra et al., 2009*; *Faucherre et al., 2009*; *Wibowo et al., 2011*; *Namdaran et al., 2012*; *Steiner et al., 2014*; *Jiang et al., 2014*). However, the regeneration of non-sensory cells remains largely unexplored. To obtain quantitative data of whole sensory-organ regeneration we developed an experimental assay that combines controllable neuromast damage, long-term videomicroscopy at cellular resolution, and live cell-lineage tracing. We used combinations of transgenic lines expressing genetically encoded fluorescent proteins that allow the precise quantification and localization of each cell class in neuromasts, and which also serve as a direct and dynamic readout of tissue organization. This is important because it enables the visualization of cell-fate transitions in living specimens within the growing tissue at high temporal resolution. Specifically, the *Tg[alpl:mCherry]* line expresses cytosolic mCherry in the mantle and interneuromast cells (*Figure 1D*). The *Et(krt4:EGFP)sqgw57A* (hereafter SqGw57A) expresses cytosolic GFP in sustentacular cells (*Figure 1E*). The *Tg[−8.0cldnb:LY-EGFP]* (Cldnb:lynGFP) express a plasma-membrane targeted EGFP in the entire neuromast epithelium and in the interneuromast cells (*Figure 1F*), and the *Tg[Sox2-2a-sfGFP]* (Sox2:GFP) expresses cytosolic GFP in all the supporting cells and the interneuromast cells (*Figure 1G*). For hair cells, we use *Et(krt4:EGFP)sqet4*(SqEt4) that expresses cytosolic GFP (*Figure 1H*), or the *Tg(myo6b:actb1-EGFP)*(Myo6b:actin-GFP) that labels filamentous actin (*Figure 1I*). These transgenic lines have been previously published, but are reproduced here for clarity and self-containment of this work (*López-Schier and Hudspeth, 2006*; *Kondrychyn et al., 2011*; *Kindt et al., 2012*; *Shin et al., 2014*; *Steiner et al., 2014*; *Pinto-Teixeira et al., 2015*).

To induce tissue damage in a controllable and reproducible manner, we used a nanosecond ultraviolet laser beam that was delivered to individual cells through a high numerical-aperture objective, which was also used for imaging. The stereotypic localization of the neuromasts along the zebrafish larva varies only marginally between individuals and during larval growth (*Figure 1J*) (*Ledent, 2002*; *López-Schier et al., 2004*). This permits the unambiguous identification of the manipulated neuromast throughout the experiment, and the comparison between corresponding organs in different animals. Using Sox2:GFP 5 day-old zebrafish larvæ that ubiquitously express a nucleus-targeted red-fluorescent protein (H2B-RFP) (*Figure 1K–L*), we certified that laser-targeted cells are rapidly eliminated from the neuromast epithelium with no detectable collateral damage (*Figure 1M–P* and *Video 1*). Having established a well-controlled injury protocol, we decided to probe the limits of neuromast regeneration. We first used specimens co-expressing Alpl:mCherry and Cldnb:lynGFP, which reveal all neuromast cells in green and the mantle cells in red (*Figure 2A*). We began by ablating entire neuromasts and assessed regeneration for 7 days (*Figure 2B–E*). Specifically, we looked at the response of flanking interneuromast cells because it has been demonstrated that they can proliferate and generate additional neuromasts, particularly upon loss of ErbB2 signaling (*López-Schier and Hudspeth, 2005*; *Grant et al., 2005*; *Sánchez et al., 2016*). Four hours post-injury (4 hpi) a wound remains evident at the target area (*Figure 2B*). One day post-injury (1 dpi), the damaged area was occupied by a thread of Alpl:mCherry(+) cells, which based on marker expression are

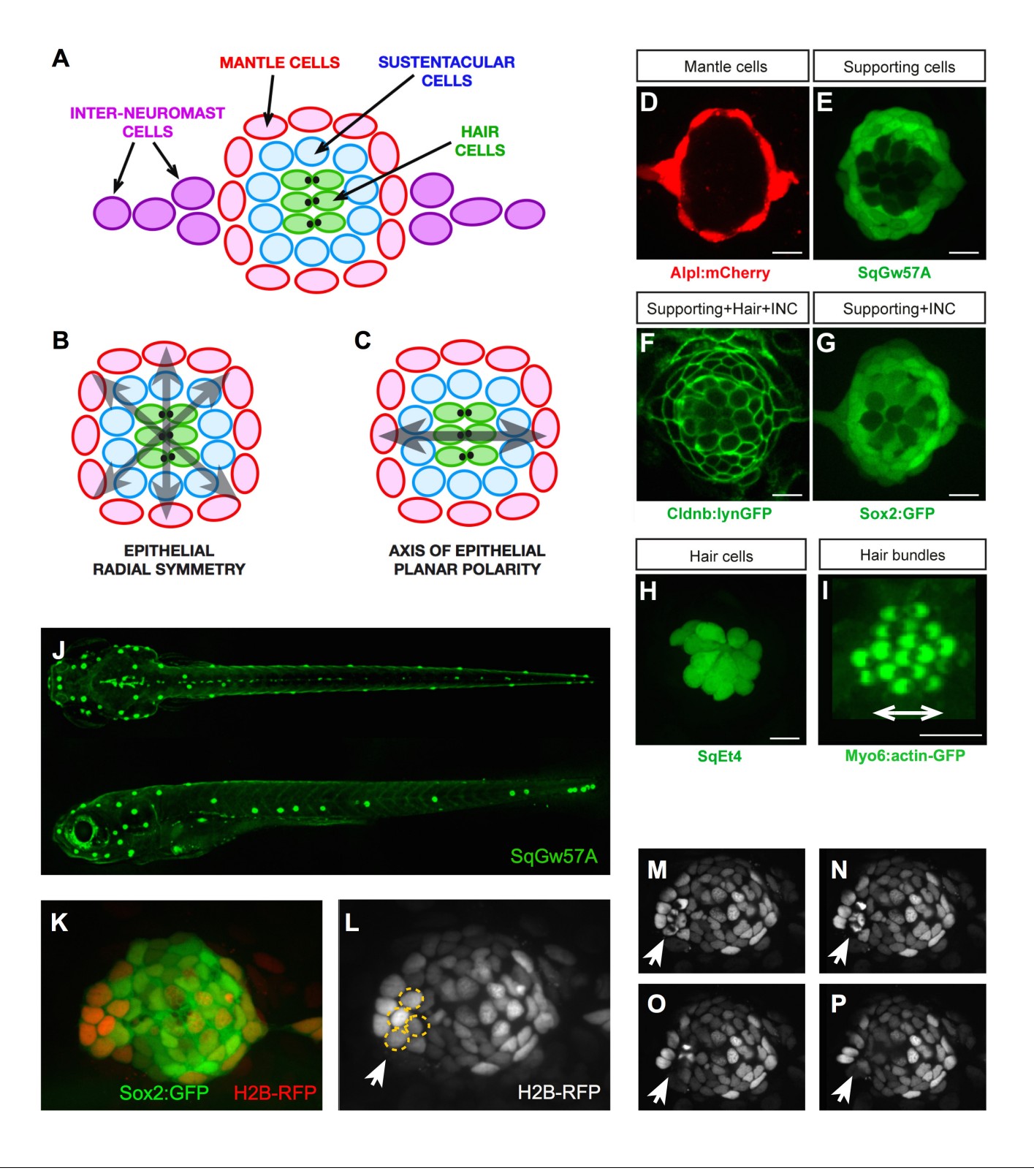

**Figure 1.** Geometric organization of the neuromast. (**A–C**) Schematic representation of a neuromast depicting (**A**) cell classes identifiable by expression of transgenic markers. Grey arrows indicate, respectively, (**B**) radial symmetry and (**C**) epithelial planar polarity. (**D–I**) Confocal images of cell-specific transgenic markers. (**D**) Alpl:mCherry marks mantle and interneuromast cells, (**E**) SqGw57A shows all supporting cells, (**F**) Cldnb:lynGFP marks all neuromast cells, (**G**) Sox2-GFP marks supporting and interneuromast cells, (**H**) SqET4 labels hair cells, and (**I**) Myo6b:actin-GFP highlights the planar

*Figure 1 continued on next page*

*Figure 1 continued*
polarization of the hair cells by decorating their apical stereocilia. Scale bars: 10 μm. (J) Images of dorsal (top) and lateral (bottom) views of a SqGw57A transgenic zebrafish larva, revealing the full complement of superficial neuromasts and their stereotypic position. (K) A single confocal section of the lateral view of a neuromast expressing GFP in supporting cells (Sox2-GFP) and a RFP in all nuclei (H2B-RFP). (L) Same neuromast in K showing RFP-marked nuclei. The white arrow indicates 4 cells (circled), which are target of the laser beam for ablation. (M–P) Four still images of the neuromast in L over a period of five minutes, in which the laser-targeted cells are eliminated from the epithelium (white arrow).
DOI: https://doi.org/10.7554/eLife.30823.002

likely interneuromast cells (*Figure 2C*). None of the removed neuromasts regenerated after 7 days (n = 22) (*Figure 2D–E*). We obtained an identical outcome using the independent pan-supporting cell marker Sox2:GFP (n = 9) (*Figure 2F–J*). Finally, incubation of Alpl:mCherry specimens with Bromodeoxy-Uridine (BrdU) to reveal the DNA synthesis that occurs prior to mitosis showed that interneuromast cells do not proliferate after neuromast ablation (*Figure 2K–N*) (*Gratzner, 1982*). These data indicate that in contrast to what occurs in embryos (*Sánchez et al., 2016*), the complete elimination of a neuromast is irreversible in larval zebrafish.

## Neuromasts have isotropic regenerative capacity

To further explore neuromast repair we decided to use milder injury regimes. We systematically produced controlled damage of well-defined scale and location in double transgenic specimens that combine the supporting cell marker Cldnb:lynGFP and the mantle-cell marker Alpl:mCherry (*Figure 3A–O*). We found that ablation of the posterior half of the neuromast was followed by closure of the wound within 24 hr (*Figure 3A–C*). At 3 dpi, target neuromasts regained normal cell-class spatial distribution (n = 6) (*Figure 3D*). At 7 dpi, neuromasts recovered approximately 70% of the normal cell number (*Figure 3E,Z*). We found no difference in speed and extent of regeneration after concurrently ablating the posterior half of neuromasts and flanking interneuromast cells (n = 5) (*Figure 3F–J,Z*). The ablation of the posterior or the dorsal half of the epithelium resulted in identical outcome, suggesting that neuromasts are symmetric in their regenerative capacity (n = 6) (*Figure 3K–O,Z*). Next, we assessed mantle-cell regeneration using a double transgenic line expressing Sox2:GFP and Alpl:mCherry, which reveal mantle cells in red and sustentacular cells in green (*Figure 3P–Y*). The complete elimination of mantle cells was followed by their re-emergence 3 dpi (*Figure 3Q–S*), and the reconstitution of the outer rim of the neuromast 7 dpi (n = 15) (*Figure 3T,Z*). The simultaneous ablation of the mantle cells and the adjacent interneuromast cells led to identical outcome (n = 6) (*Figure 3U–Z*). The ablation of the interneuromast cells in fish co-expressing Sox2:GFP and Alpl:mCherry on one side of a neuromast (n = 12), or between two adjacent organs (n = 8) did not trigger the proliferation of the remaining interneuromast cells over a period of 7 days (*Figure 3—figure supplement 1A–J*). Because the complete ablation of mantle cells leaves intact the sustentacular-cell population, and the hair cells are postmitotic, these results yield three important and novel findings: (1) interneuromast cells are not essential for neuromast regeneration in larval zebrafish, although they may contribute to mantle cell regeneration; (2) neuromasts have isotropic regenerative capacity; (3) sustentacular cells are tri-potent progenitors able to self-renew and to generate mantle and hair cells.

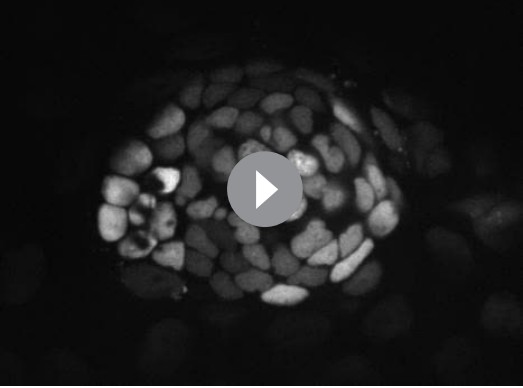

**Video 1.** A 20 min videomicroscopic recording of a neuromast after laser-mediated ablation of supporting cells. Four laser-targeted cells (showing a dark spot in the nuclei from focal fluorescent-protein bleaching) are eliminated from the epithelium, which closes the wound. There is no noticeable collateral damage. Time resolution is one image per 30 s.
DOI: https://doi.org/10.7554/eLife.30823.004

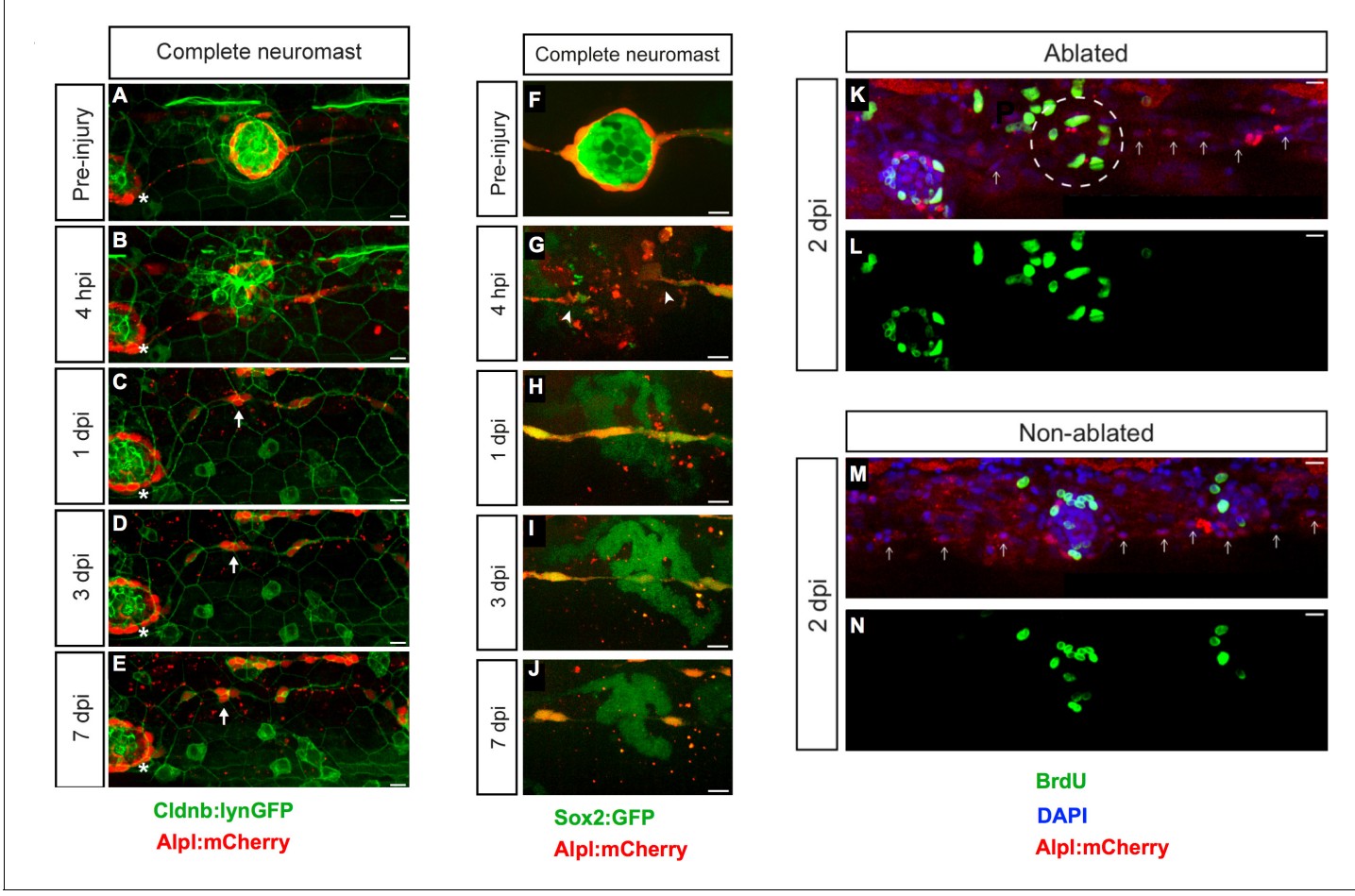

**Figure 2.** Zebrafish larvæ do not regenerate completely-ablated neuromasts. (A–E) Confocal images of a 7 day follow-up of the complete ablation of a neuromast in the double transgenic line *Tg[Cldnb:lynGFP; Alpl:mCherry]*. (A) The site of damage was identified over subsequent days by the position of an intact reference neuromast (white asterisk). (B) Laser-mediated cell ablation produced a wound 4 hours-post-injury (hpi). (C–E) This wound was replaced by a thread of mCherry(+) cells (white arrow) 1 day-post-injury (dpi), which did not change over the subsequent 6 days. (F–J) Confocal images over a 7 day time course after the ablation of a neuromast in the double transgenic line *Tg[Sox2:GFP; Alpl:mCherry]*. Identically to A-E, the complete ablation of the target neuromast results in a thin trail of interneuromast cells (white arrowheads) covering the damaged area (K–N). Scale bars: 10 μm.
DOI: https://doi.org/10.7554/eLife.30823.003

## Neuromast architecture recovers after severe loss of tissue integrity

To test the limits of neuromast regeneration we systematically ablated increasing numbers of cells. Extreme injuries that eliminated all except 1 to 3 cells almost always led to neuromast loss (not shown), whereas ablations that left between 4 and 10 cells, reducing the organ to a combination of 2–3 mantle and 2–7 sustentacular cells, allowed regeneration (*Figure 4A–E,K*). We found that after losing over 95% of their cellular content, neuromasts recover an average of 45 cells at 7 dpi (or approximately 70% of the normal cell count), with exceptional cases reaching 60 cells (equivalent to over 90% of a normal organ) (n = 15) (*Figure 4K*). Regenerating neuromasts became radial-symmetric as early as 3 dpi (*Figure 4D*), and had normal cell-class composition and proportions 7 dpi (*Figure 4L–M*). Next, we concurrently ablated 95% of the neuromast and the flanking interneuromast cells (*Figure 4F–G*). This intervention was followed by a similar regeneration process, but lead to smaller organs (n = 6) (*Figure 4H–J,N–P*). These observations reinforce our previous suggestion that interneuromast cells have a non-essential, yet appreciable contribution to regeneration. Timed quantification of cell-class number and localization showed a reproducible pattern of tissue growth and morphogenesis. During the first 24 hpi, the intact cells rebuilt a circular epithelium (*Figure 4B*). From 1 dpi to 3 dpi, cell number increases rapidly and proportion is restored (*Figure 4C,K–M*). After

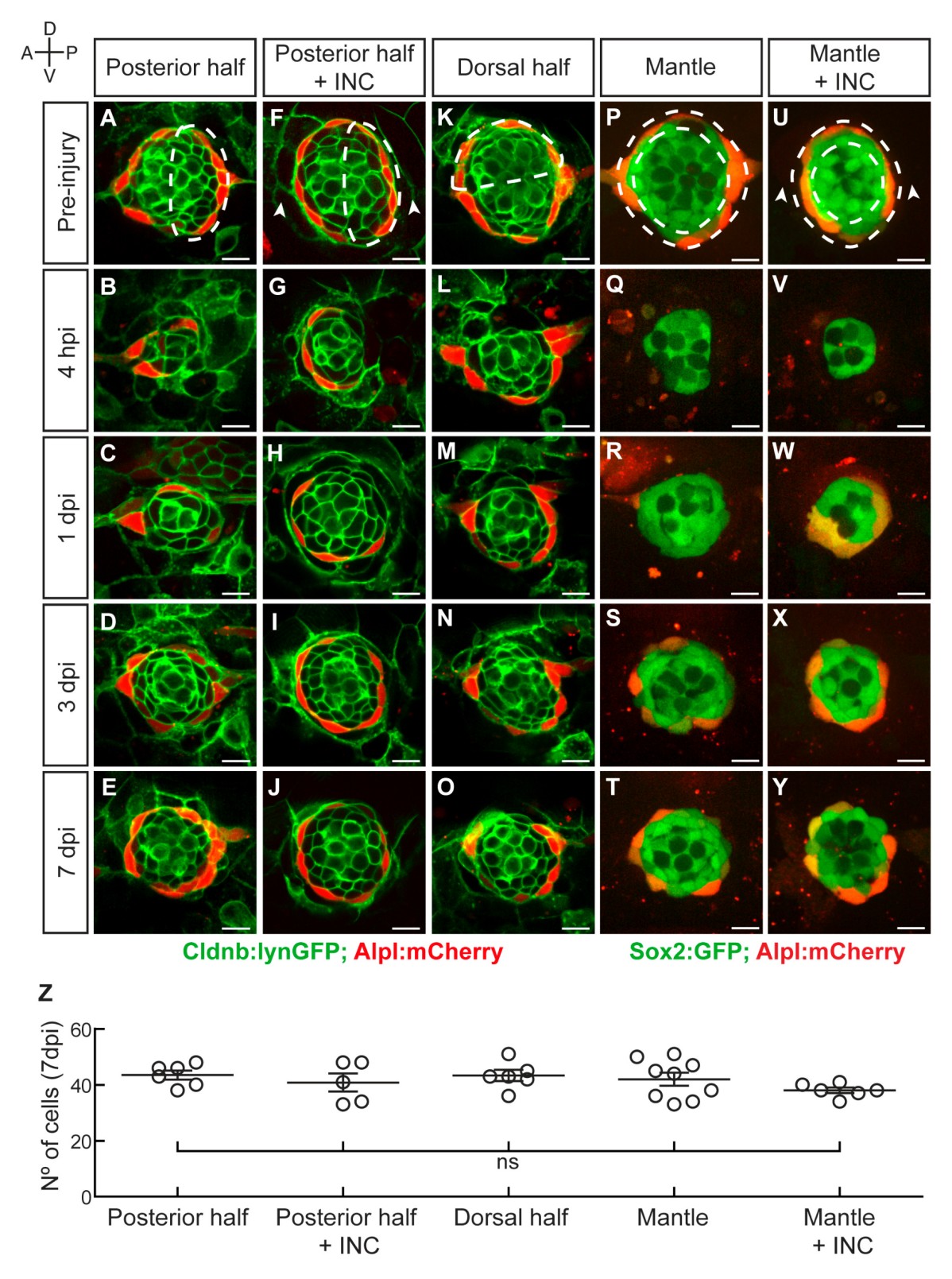

**Cldnb:lynGFP; Alpl:mCherry**

**Sox2:GFP; Alpl:mCherry**

**Figure 3.** Neuromasts have isotropic regenerative capacity. (A) Ablation of the posterior half of a neuromast. (B–C) The damage is resolved by cellular movement from the undamaged site 1dpi. (D) Neuromasts recover geometric order after 3 days and (J) return to homeostasis by 7dpi. Dashed lines in A,F,K,P,U delineate the ablated area. (F–J) Simultaneous ablation of the posterior half of a neuromast and the interneuromast cells flanking its anterior and posterior sides (n = 5) led to a regeneration outcome identical to that of the experiment in (A–E). Arrowheads in (F) point the location normally

*Figure 3 continued on next page*

*Figure 3 continued*

occupied by the interneuromast cells. (**K–O**) Neuromasts depleted from their dorsal half (n = 6) also recover epithelial size, proportions and geometry in a manner indistinguishable from equatorial-side ablation after 7 days. (**P–T**) 7 days after their complete laser-mediated ablation, mantle cells regenerated for neuromasts to recover the mantle. (**U–Y**) The ablation of interneuromast cells flanking both sides of neuromasts that were depleted of mantle cells resulted in the same outcome (n = 6). (**Z**) Quantification of the number of cells in regenerated neuromasts at 7 dpi. Number of neuromast cells was no statistically significant between groups of different damage regimes as determined by one-way ANOVA (F(4,27)=1.013, p=0.4183). Scatter plot shows mean ±s.e.m.ns: non-significant. Scale bars: 10 μm.

DOI: https://doi.org/10.7554/eLife.30823.005

The following figure supplement is available for figure 3:

**Figure supplement 1.** Interneuromast cells do not regenerate.

DOI: https://doi.org/10.7554/eLife.30823.006

3 dpi, cell number increases at a slower pace (*Figure 4K–M*). Importantly, each cell class assumes an appropriate position despite a much reduced cell number (*Figure 4E,J,L–P*).

Next, we examined if the orthogonal polarity axes of the epithelium are re-established after the severest of injuries. To assess tissue apicobasal polarity we used a combination of transgenic lines that allows the observation of the invariant basal position of the nucleus and the apical adherens junctions (*Figure 4Q–R*) (*Ernst et al., 2012*; *Harding and Nechiporuk, 2012*; *Hava et al., 2009*). We found correct positioning of these markers in the regenerated epithelium (n = 4), including the typical apicobasal constriction of the hair cells (*Figure 4S–T*). To assess epithelial planar polarity, we looked at hair-bundle orientation using fluorescent phalloidin, which revealed that 7 dpi the regenerated neuromasts were plane-polarized in a manner indistinguishable from unperturbed organs, with half of the hair cells coherently oriented in opposition to the other half (n = 10) (*Figure 4U–W*). To test if plane-polarizing cues derive from an isotropic forces exerted by the interneuromast cells that are always aligned to the axis of planar polarity of the neuromast epithelium, we ablated these cells flanking an identified neuromast, and concurrently killed the hair cells with the antibiotic neomycin (*Figure 4X–Y*). In the absence of interneuromast cells regenerating hair cells recovered normal coherent planar polarity (n = 16), suggesting the existence of alternative sources of polarizing cues (*Figure 4Z*). Collectively, these findings reveal that as few as four supporting cells can initiate and sustain integral organ regeneration.

## Sustentacular and mantle cells have different regenerative potential

Injury in the wild is intrinsically stochastic. Thus, we hypothesized that the regenerative response must vary according to damage severity and location, but progress in a predictable manner. To test this assumption and unveil the underlying cellular mechanism, we systematically quantified the behavior of individual cells by high-resolution videomicroscopy. We conducted 15 independent three-dimensional time-lapse recordings of the regenerative process using a triple-transgenic line co-expressing Cldnb:lynGFP, SqGw57A and Alpl:mCherry (*Figure 5A–B*), ranging from 65 to 100 hr of continuous imaging (each time point 15 min apart). Starting immediately after the ablation of all except 4–10 cells, we tracked every intact original cell (called founder cell) and their progeny (cellular clones) (*Figure 5A* and *Video 2*). We followed a total 106 founder cells (76 sustentacular cells and 30 mantle cells). We tracked individual cells manually in space and time, recording divisions and identity until the end of the recording, resulting in 763 tracks and 104,863 spatiotemporal cell coordinates (*Figure 5A–B*). Each clone was represented as a tree to visualize the contribution of each founder cell to the resulting clones (*Figure 5C*). We found that the majority of the founder sustentacular cells underwent three divisions and that some divided up to five times (*Figure 5D*). 14 out of 30 founder mantle cells did not divide at all, and the rest divided once or, rarely, twice. Founder sustentacular cells required on average 19 ± 6 hr (mean ±s.d., n = 76) to divide, whereas the founder mantle cells that divided required on average 27 ± 5 hr, (mean ±s.d., n = 30) (*Figure 5E*). Clones from founder sustentacular and founder mantle cells were markedly different: founder sustentacular cells produced all three cell classes (sustentacular, mantle and hair cells), whereas founder mantle cells produced clones containing only mantle cells (*Figure 5F*). We categorized all cell divisions according to the fate of the two daughter cells at the time of the following division, or at the end of the time-lapse recording (*Figure 5G*). This analysis revealed that 97% of the sustentacular-cell divisions were symmetric: 78% produced two sustentacular cells (SS), 16% produced a pair of hair cells (HH), and

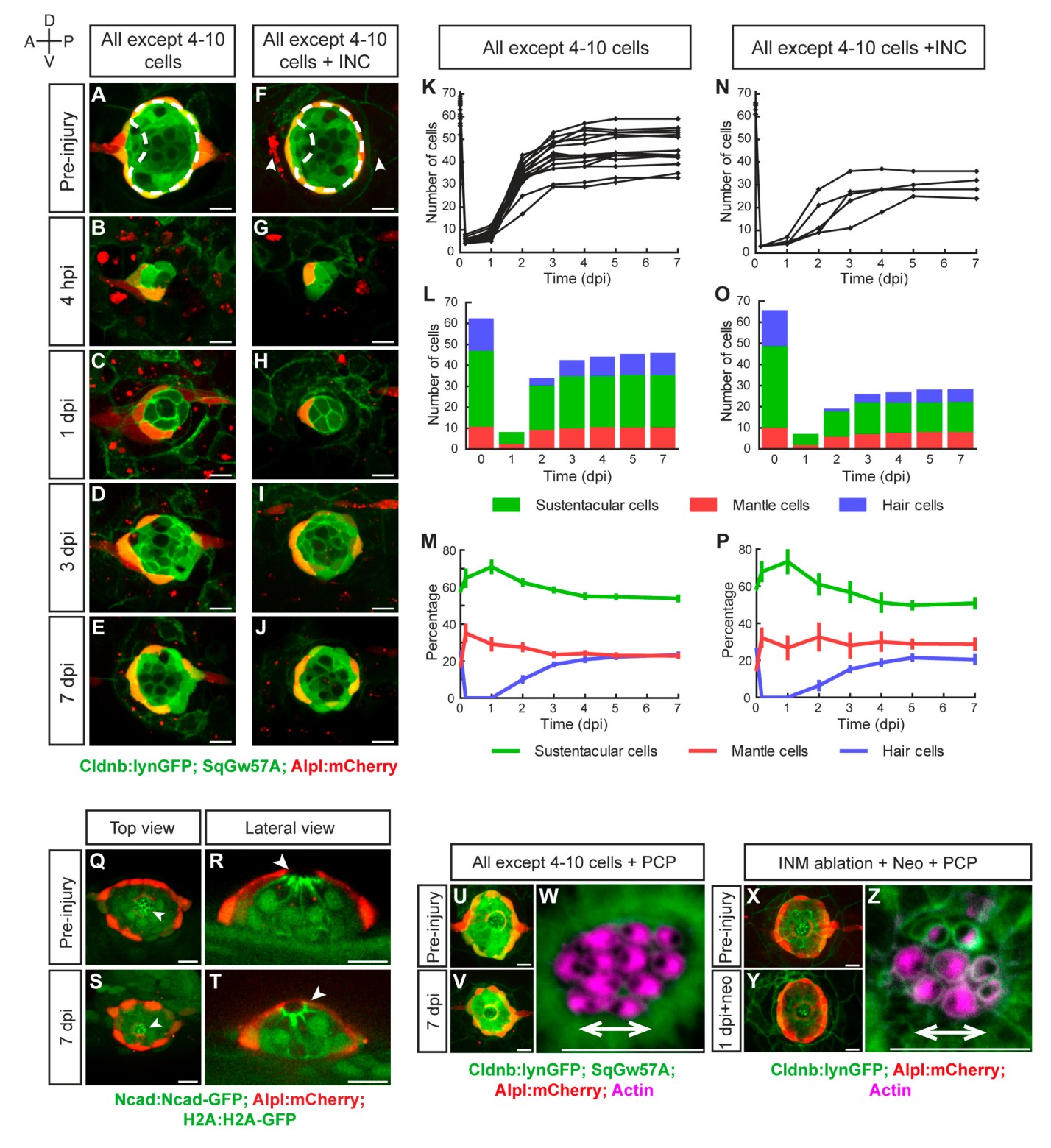

**Figure 4.** Recovery of organ architecture after loss of tissue integrity. (A–E) Confocal images of a neuromast regenerating from 4 to 10 cells during a period of 7 days. Neuromasts recover radial symmetry 3 dpi (D), and original organ proportions at 7 dpi (E). (F–J) Neuromasts reduced to 4–10 cells that were previously deprived from adjacent interneuromast cells (INCs) (arrowheads in F), regenerated and reformed radial symmetry (H–I) and proportions 7 dpi, despite maintaining a reduced size (J). Dashed circles in (A,F) illustrate damaged areas. Scale bars: 10 µm. (K,N) Total cell numbers in regenerating neuromasts over 7 days in the two conditions depicted in (A–J). (L,O) In the first 2 dpi neuromast consist almost exclusively of

*Figure 4 continued on next page*

*Figure 4 continued*

supporting cells (green and red). Hair cells (blue) begin to appear between at 2dpi. (M,P) Percentages of cell classes during a 7 day regeneration period. Right after damage, neuromast experience an imbalance of cell proportions that is re-established over the course of 3 days. Afterwards the neuromasts continues to slowly increase total cell number at similar rates. The final proportion of cell classes recapitulates that of the starting condition. Time points show mean ± s.e.m. [All except 4–10 cells] n = 15, [All except 4–10 cells + INC] n = 6. (Q) Top and (R) lateral views of a triple-transgenic *Tg [Ncad: Ncad-EGFP; Alp:mCherry; H2A:H2A-EGFP]* neuromast before injury. (S) Top and (T) lateral views of a regenerated neuromast 7 days post injury (n = 4). Basal location of nuclei and apical N-cadherin enrichment evidence the apicobasal polarization of the organ. The accumulation of N-cadherin (white arrowheads) in the regenerated neuromast shows that apical constrictions are properly re-established during the process. (U–V) Maximal intensity projection of a neuromast in the triple transgenic line *Tg[Cldnb:lynGFP; SqGw57A; Alpl:mCherry]* prior to injury that eliminates all except 4 to 10 cells (U), and the same neuromast 7 days after damage (V). (W) Hair-bundle staining with rhodamine-phalloidin (colored in pink) reveals the coherent planar polarization of the hair cells in the regenerated neuromast shown in (V). (X) Confocal projection of a neuromast before the removal of flanking interneuromast cells. (Y) Maximal projection of a neuromast 48 hr after interneuromast-cell ablation and 24 hr after neomycin treatment. (Z) Phalloidin staining of hair bundles of hair cells regenerated in the absence of interneuromast cells, showing recovery of coherent epithelial planar polarity. Scale bars: 10 μm.

DOI: https://doi.org/10.7554/eLife.30823.007

3% generated two mantle cells (MM). Only 3% of the divisions were asymmetric, generating one sustentacular and one mantle cell (SM) (n = 307). All mantle-cell divisions were symmetric (MM) (n = 20). These observations further support the conclusion that sustentacular cells are tri-potent progenitors.

Previous studies have firmly established that hair-cell regeneration is strongly anisotropic because hair-cell progenitors develop almost exclusively in the polar areas of horizontal neuromasts, elongating the macula in the dorsoventral direction (*Wibowo et al., 2011*; *Romero-Carvajal et al., 2015*). Although our static images suggest that neuromasts have isotropic regenerative capacity, we nevertheless wondered whether regeneration of non-sensory cells is directional. To this end, we fractioned the epithelium of horizontal neuromasts in four quarters of equal dimension (dorsal, ventral, anterior and posterior) (*Figure 6A–B*), which reflects the known functional territorialization of the neuromast epithelium based on the expression of transgenic markers and Notch signaling (*Ma et al., 2008*; *Wibowo et al., 2011*). We first assessed the spatial distribution of cell divisions during the first 60 hr of regeneration and found no pattern that would suggest regeneration anisotropy (*Figure 6A*). However, 60 hpi, most divisions (74%) took place in the dorsal and ventral (polar) quarters (*Figure 6B*). This is expected because later divisions mainly produce hair cells from polar progenitors (*Figure 4L, M*). Thus, the regenerating epithelium is initially homogeneous and becomes territorialized 60 hpi. We reasoned that epithelial territorialization could occur either by the migration of similar cells that are scattered throughout the tissue, or by position-adaptive differentiation of an initially equivalent population of cells. To test these possibilities, we generated a virtual Cartesian coordinate system at the center of the neuromast to fit all founder cells at the beginning of regeneration (4hpi). Next, we analyzed the localization of their progeny 60 hpi (*Figure 6C–H*). We found that 60% of the progeny of anterior-localized founder cells were located in the anterior side of the resulting epithelium, whereas 64% of the progeny of posterior-located founder cells were found in the posterior side (*Figure 6C–E*). We also found that 72% of cells derived from dorsal founder cells and 74% of cells from ventral founder cells were located on the same side of the virtual dorsal/ventral midline (*Figure 6F–H*). Therefore, most of the clones remain ipsilateral to the founder cell. These results indicate that neuromasts have isotropic regenerative capacity and their territorialization occurs by location-adaptive cellular differentiation.

## The sustentacular-cell population is tri-potent and plastic

To answer the long-standing question of whether the sustentacular-cell population is homogeneous and approach the problem of what determines symmetric versus asymmetric modes of division, we characterized the composition of all 72 clones from founder sustentacular cells. We found four types of clones: containing only sustentacular cells (S), sustentacular and mantle cells (SM), sustentacular and hair cells (SH), and all three cell classes (SHM) (*Figure 6I*). Of note, founder mantle cells produced clones containing only mantle cells (M) (*Figures 5G* and *6I*). We observed that 37/72 of the clones from founder sustentacular cells were SH, 21/72 were S, 12/72 were SM, and 2/72 were SHM (*Figure 6I–K*). The proportion of each clone type suggests that either the sustentacular-cell

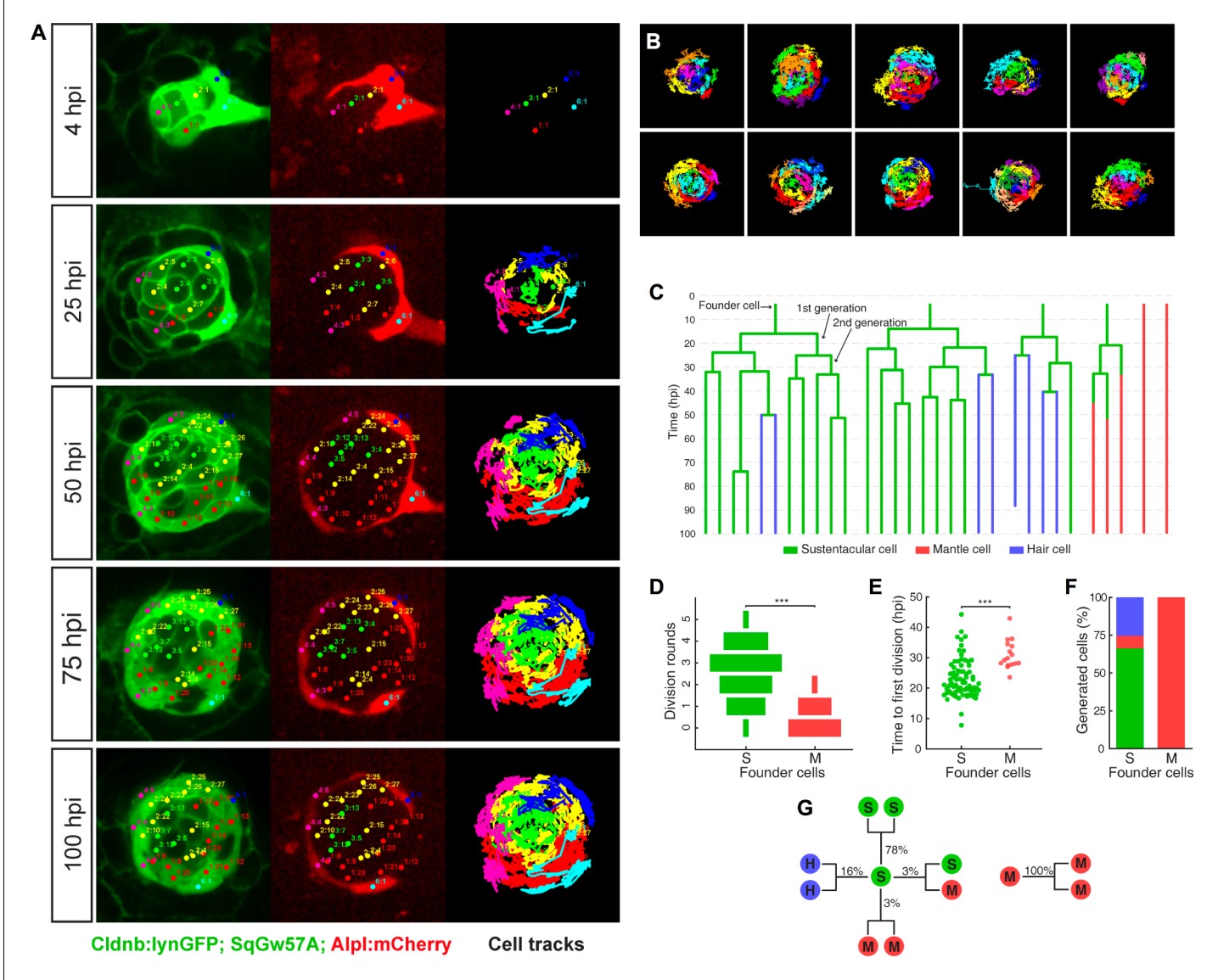

**Figure 5.** Long-term whole-organ single-cell tracking reveals cell-clone formation during neuromast regeneration. (A) Still images showing a representative 100 hr time-lapse recording of a regenerating neuromast in *Tg[Clndb:lynGFP; SqGw57A; Alpl:mCherry]* larva (left and middle panels). Cellular clones that share a common founder cell are clustered and color-coded. Cell trajectories reveal a concentric growth pattern (right panel). (B) Cell trackings at the last recorded timepoints for 10 out of the total of 15 regenerated neuromasts. (C) Cell-lineage tracing from time-lapse movie shown in (A). Branching points symbolize cell divisions. The division of a founder cell generates two cells of the 1 st generation. Subsequent divisions produce cells of the 2nd, 3rd and 4th generation. Cell classes are indicated with green (sustentacular), blue (hair) and red (mantle) colors. (D) Sustentacular founder cells undergo significantly more (p=3.59e-06, Mann-Whitney test) division rounds than mantle founder cells during 100 hr of neuromast regeneration. (E) The first division of sustentacular founder cells (n = 76) occurs significantly earlier (p=1.13e-5, Mann-Whitney test) than that of mantle founder cells (n = 16). (F) Sustentacular founder cells (n = 76) generate all three neuromast cell classes whereas mantle founder cells (n = 30) produce only mantle cells. (G) Out of 307 sustentacular cell divisions, 78% were self-renewing, 16% produced a pair of hair cells, 3% produced sustentacular cells that both became mantle cells within the next generation and 3% generated two sustentacular cells of which only one transited to mantle cell fate within the next generation. All 20 observed mantle cell divisions were self-renewing.

DOI: https://doi.org/10.7554/eLife.30823.008

population is heterogeneous, or that it is homogeneous but plastic. In searching for potential sources of clone heterogeneity, we noted that in some developmental contexts cell-cycle length or proliferative potential can determine the fate of the daughter cells (*Calegari et al., 2005*; *Rossi et al., 2017*). Therefore, we quantified the kinetics of proliferation of founder sustentacular cells and of

their daughters and compared them to clone composition. We found three clear waves of cell divisions, each spaced by 8–10 hr (*Figure 7A*), respectively peaking at 20 hr, 28 hr and 38 hr (*Figure 7B–C*), suggesting that cell-cycle length is strictly regulated. Cell-cycle length in the 1 st generation peaks around 10 hr (9.8 ± 3.3 hr, median ±interquartile range (iqr)) (*Figure 7C*), but it begins to increase and to vary in the 2nd generation (11.5 ± 7.3 hr, median ±iqr), and more so in the 3rd generation (18.8 ± 20.3 hr, median ±iqr). To identify transition points in cycle lengths, we tested the goodness of fit of a two-segment regression model with variable change points. We found that the length of cell cycles is initially around 11 ± 3 hr (mean ±s.d.) and slowly increases up to 47 hpi. Afterwards, cell-cycle length increases more rapidly and is more variable (*Figure 7D*). To test if cell number influences cell-cycle length we used a similar two-segment regression model to define when cell-cycle length loosens, and discovered that the vast majority of the cell cycles (76%) span 7–13 hr below a threshold of 24 cells (*Figure 7E*). Above this threshold, cell-cycles lengths show large variation. With these data, we plotted proliferation kinetics against clone type, and found no significant difference between clones (*Figure 7F–G*). Thus, the length of the cell cycle or the proliferative potential of founder sustentacular cells cannot explain clone composition.

## Machine learning identifies predictive features for cell-fate acquisition

Multiple extrinsic factors that vary in space and time could determine cell-fate choices. Because manual analysis of such multidimensional data might be biased or neglect certain factors we implemented a quantitative and unbiased computational approach based on machine learning to identify variables (features) that correlate with clone composition. The first step of the workflow is the extraction of spatiotemporal coordinates and cell-lineage information from the manual tracks of the video-microscopic data sets (n = 15) (*Figure 8A*). For each cell-track coordinate, we extracted 32 quantifiable features (*Table 1*), which were used to train the machine-learning algorithm. In a pre-analysis, we compared the performance of 20 algorithms (support vector machines, decision trees and nearest neighbor classifiers) in terms of accuracy and area under the curve (AUC) and chose the ensemble bagged tree random forest algorithm (*Breiman, 2001*) as the best performing method (*Figure 8—figure supplement 1*). To avoid overfitting, we trained the random forest using 14 samples to predict clone composition in the remaining sample in a round robin fashion. We evaluated the quality of predictions using Matthews correlation coefficient (MCC) to compensate for imbalances of clone frequencies (*Figure 6K*)

Using machine learning, we were able to predict the occurrence of SH vs. SM clones from a founder sustentacular cell with high accuracy (42 out of 49 correctly predicted clones, MCC = 0.63 ± 0.09, mean ± s.d., n = 15 bootstrapped samples), while neither SH nor SM clones could be discriminated when compared to S clones (*Figure 8B*). Of the 32 features that we used, those that best discriminated SH vs SM clones were the sustentacular cells' distance to the center of the epithelium, and the distance to the mantle cells (*Figure 8C* and *Figure 8—figure supplement 2*). Next, we focused on the decision-making process of individual sustentacular cells at the time of their division. We trained a random forest to discriminate between SS, HH and SM/MM divisions in a pairwise fashion. The HH and SM/MM divisions were highly predictable (63 out of 66 divisions correctly predicted, MCC = 0.91 ± 0.07, mean ± s.d., n = 15 bootstrapped samples), while the discrimination between SS and HH or SM/MM divisions was much less accurate (*Figure 8D*). Again, the most informative features were the distance to the neuromast center and the distance to the mantle cells (*Figure 8E*, *Figure 8—figure supplement 3*). SM/MM divisions occur consistently at the outer perimeter of the neuromast (*Figure 8F*), whereas HH divisions take place near the center. Self-renewing SS divisions occupy the area between HH and SM/MM divisions. Interestingly, SM/MM divisions were never seen in the anterior-most region of the organ, suggesting that progenitor sustentacular

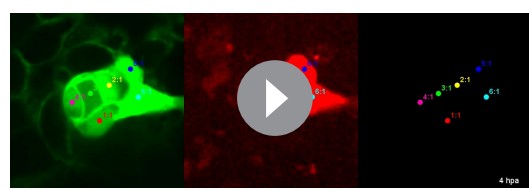

**Video 2.** 100 hr time-lapse recording of a regenerating neuromast after severe ablation. A neuromast regenerates its original architecture from as few as six founder cells. Founder cells are identified by 1–6 (*n*) and their daughter cells receive *2 n* and *2n + 1* identities. Recording starts 4 hr post injury (hpi) and shows single focal planes. Time is in hours post injury.
DOI: https://doi.org/10.7554/eLife.30823.009

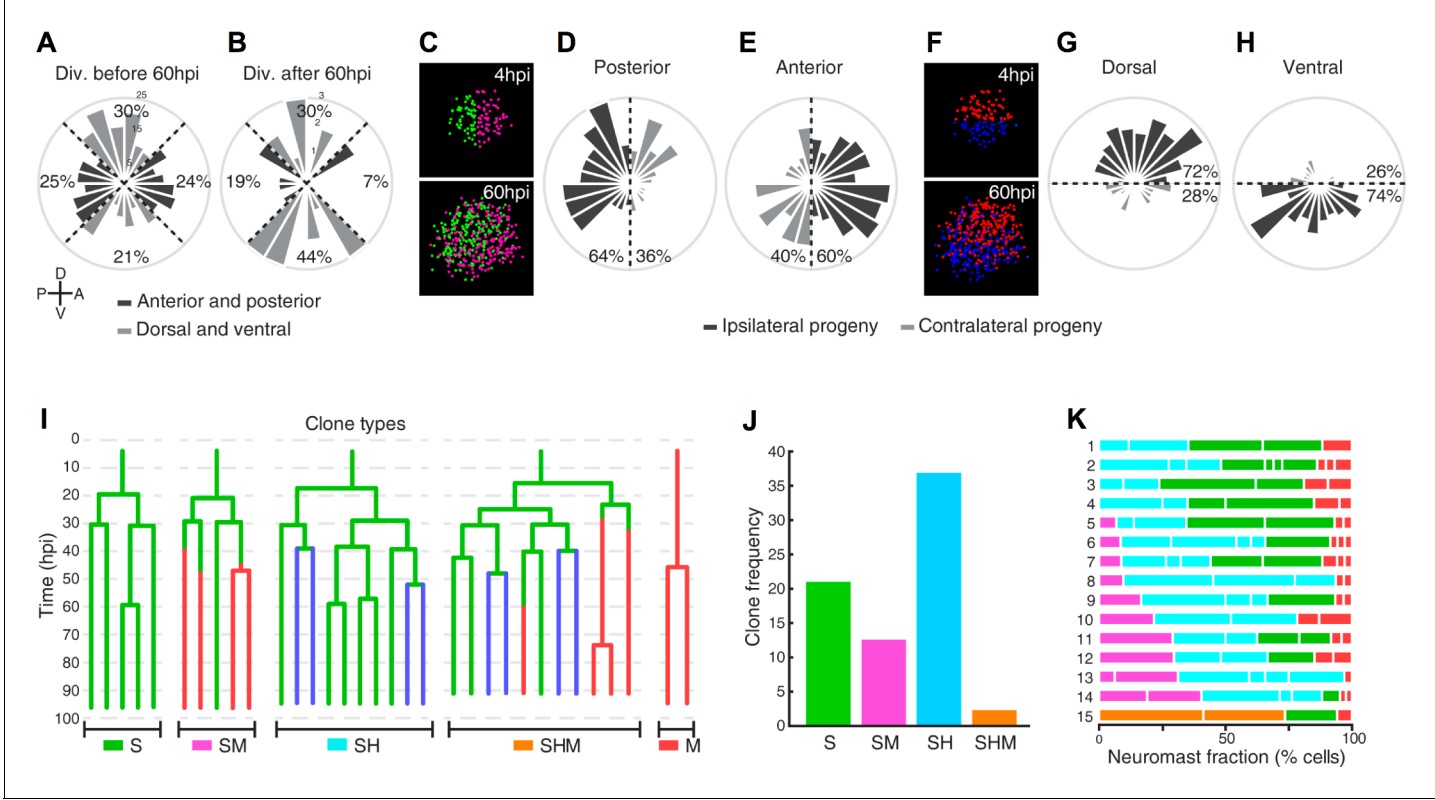

**Figure 6.** Neuromast regeneration is not stereotypic and reveals different clone type compositions. (**A**) Proliferation is markedly isotropic during the first 60 hr of neuromast regeneration (n = 348). (**B**) Homeostatic, dorso-ventral (DV) proliferative territories are restored after 60hpi (n = 27). (**C**) 40% and 36% of the progeny from anterior (n = 202) and posterior (n = 173) founder cells crossed to the contralateral side (light grey) after 60 hr of regeneration. (**D**) Only 28% and 26% of the progeny from dorsal (n = 199) and ventral (n = 176) founder cells crossed to the contralateral side (light grey) during the same period of time. (**E**) Representative examples of different clone types extracted from time-lapse data. Sustentacular cells give rise to S, SM, SH, and SHM clones (color coded respectively with green, pink, cyan and orange) whereas mantle cells produce only pure mantle cell clones. (**F**) The clone composition of the 15 regenerated neuromasts is not stereotypic. The length of each bar represents the proportion of neuromast cells that belong to each clone. Neuromast eight has been shown in *Figure 5A,B*. (**G**) The most frequent clones contain sustentacular and hair cells (SH, n = 37 clones), followed by those with only sustentacular cells (S, n = 21 clones). The third most frequent are composed by sustentacular and mantle cells (SM, n = 12 clones). Clones containing all three cell classes were rare (SHM, n = 2 clones).

DOI: https://doi.org/10.7554/eLife.30823.010

cells are routed into generating mantle cells specifically in the perimetral areas that lack mantle cells but not elsewhere. Therefore, regenerating neuromasts appear to sense cell-class composition and route cellular differentiation in a spatially regulated manner to regain cell-class proportion and distribution.

## Discussion

One long-standing goal of biological research is to understand the regeneration of tissues that are exposed to persistent environmental abrasion. Here we address this problem by developing a quantitative approach based on videomicroscopic cell tracking, cell-lineage tracing, and machine learning to identify features that predict cell-fate choices during organ regeneration. Using the superficial neuromasts in zebrafish, we demonstrate that a remarkably small group of resident cells suffices to rebuild a functional organ following severe disruption of tissue integrity. Our findings reveal that the sustentacular-cell population is tri-potent, and suggest that integral organ recovery emerges from multicellular organization employing minimal extrinsic information. Below, we discuss the evidence that supports these conclusions.

By systematically analyzing cellular behavior, we reveal a hierarchical regenerative process that begins immediately after injury. First, surviving founder cells reconstitute an epithelium. Second,

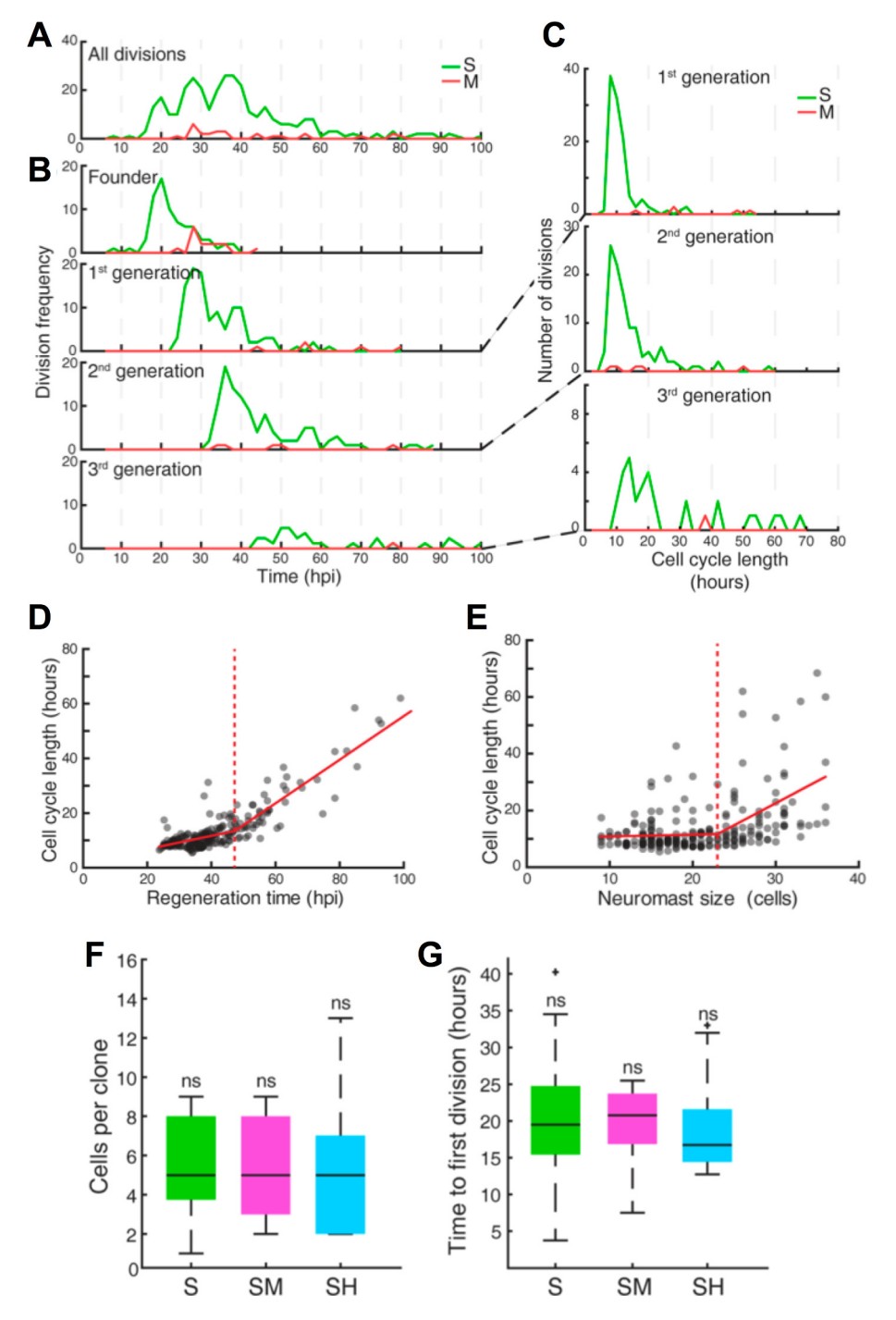

**Figure 7.** Quantification of cell divisions during neuromast regeneration. (**A–B**) Equally spaced waves of coordinated sustentacular cell divisions (green) underlie the recovery of neuromast cell size. Mantle cell divisions (red) occur occasionally and do not follow the pattern of sustentacular cells. Proliferative waves correspond to the coordinated divisions of cells from independent generations. (**C**) Cells from the 1 st and 2nd generation divide on average after cell cycles of 11 ± 5 and 14 ± 9 hr respectively (mean ±s.d.). Coordination is lost at the 3rd generation when cell cycles start to lengthen (26 ± 18 hr, mean ±s.d.). (**D**) Cell cycle length (11 ± 3 hr, mean ±s.d.) is marginally influenced by regeneration time until 47 hr after injury, when cycle length starts increasing proportionally with regeneration time. (**E**) Cell cycle lengths (12 ± 6 hr, mean ±s.d.) do not correlate directly with neuromast size until 24 neuromast cells. (**F**) S, SM and SH clones produce similar number of cells (p=0.68, Kruskal Wallis test). In the box plots, the boundary of the box indicates the 25th and 75th percentile, respectively the black
*Figure 7 continued on next page*

*Figure 7 continued*

line within the box marks the median. Whiskers above and below the box include points that are not outliers. (**G**) Sustentacular founder cells of S, SM, and SH clones divide similarly early (p=0.42, Kruskal Wallis test) after approximately 18 hr after neuromast injury. (**H**) Sustentacular founder cells that produce SH (cyan) and S clones (green) are distributed similarly around the center of the organ (at x = y = 0). Those that generate SM clones (pink) are localized further away from the center and are biased towards the posterior side.

DOI: https://doi.org/10.7554/eLife.30823.011

sustentacular cells become proliferative and restore organ size. Cellular intercalation is rare. Third, daughter cells differentiate in a position-appropriate manner to recreate cell-class proportions and organ geometric order. Fourth, the epithelium returns to a homeostatic state that is characterized by low mitotic rate. The milder damage regimes that eliminated one half of the epithelium show that neuromasts are symmetric in their regenerative capacity, and that they preferentially regenerate the cells that have been eliminated. Importantly, these findings, which rely on the quantitative spatio-temporal analysis of regeneration data, could not have been predicted from previous studies using static and largely qualitative information (*Williams and Holder, 2000*; *López-Schier and Hudspeth, 2005*; *Dufourcq et al., 2006*; *López-Schier and Hudspeth, 2006*; *Ma et al., 2008*; *Wibowo et al., 2011*; *Wada et al., 2013*; *Steiner et al., 2014*; *Romero-Carvajal et al., 2015*; *Cruz et al., 2015*; *Pinto-Teixeira et al., 2015*). An important corollary of these results is that neuromasts do not contain specialized cells that contribute dominantly to repair. We propose that progenitor behavior is a facultative status that every sustentacular cell can acquire or abandon during regeneration. We did not observe regenerative overshoot of any cell class (*Agarwala et al., 2015*), suggesting the existence of a mechanism that senses the total number of cells and the cell-class balance during tissue repair (*Simon et al., 2009*). Together with previous work, our results support the possibility that such mechanism is based on the interplay between Fgf, Notch and Wnt signaling (*Ma et al., 2008*; *Wibowo et al., 2011*; *Wada et al., 2013*; *Romero-Carvajal et al., 2015*; *Dalle Nogare and Chitnis, 2017*). Our combination of machine learning and quantitative videomicroscopy shows clear differences between sustentacular and mantle cells, but does not indicate heterogeneity within the sustentacular-cell population. However, further application of this integrated approach and new transgenic markers may reveal uncharacterized cells in the neuromast. This may be expected given recent work that showed the existence of a new cell class in neuromasts of medaka fish (*Seleit et al., 2017*). It is technically challenging to consistently maintain fewer than 4 cells *in toto* without eliminating the entire neuromast. Thus, we cannot rule out the possibility that a single founder cell may be able to regenerate a neuromast. We show that the complete elimination of a neuromast is irreversible in larval zebrafish. However, Sánchez and colleagues have previously reported that interneuromast cells can generate new neuromasts (*Sánchez et al., 2016*). By assaying DNA synthesis prior to mitosis, we show that interneuromast cells do not proliferate after neuromast ablation. These differences may be explained by differences in ablation protocols (electroablation versus laser-mediated cell killing), the age of the specimens (embryos versus early larva) or the markers used to assess cellular elimination.

We find that interneuromast cells are not essential for neuromast regeneration because severely damaged organs recover all cell classes in the appropriate localization in the absence of interneuromast cells. However, we systematically observed smaller organs when interneuromast cells where ablated. These observations suggest that these peripheral cells may yet help regeneration, either directly by contributing progeny, or by producing mitogenic signals to neuromast-resident cells.

The behavior of the mantle cells is especially intriguing. Complete elimination of parts of the lateral line by tail-fin amputation have revealed that mantle cells are able to proliferate and generate a new primordium that migrates into the regenerated fin to produce new neuromasts (*Dufourcq et al., 2006*). This observation can be interpreted as suggesting that under some injury conditions, mantle cells are capable of producing all the cell classes of a neuromast. Transcriptomic profiling of mantle cells following neuromast injury revealed that these cells up-regulate the expression of multiple genes (*Steiner et al., 2014*). Furthermore, a recent study has revealed that mantle cells constitute a quiescent pool of cells that re-enters cell cycle only in response to severe depletion of sustentacular cells (*Romero-Carvajal et al., 2015*), suggesting that these cells may conform a stem-cell niche for proliferation of sustentacular cells. Thus, the collective evidence indicates that the

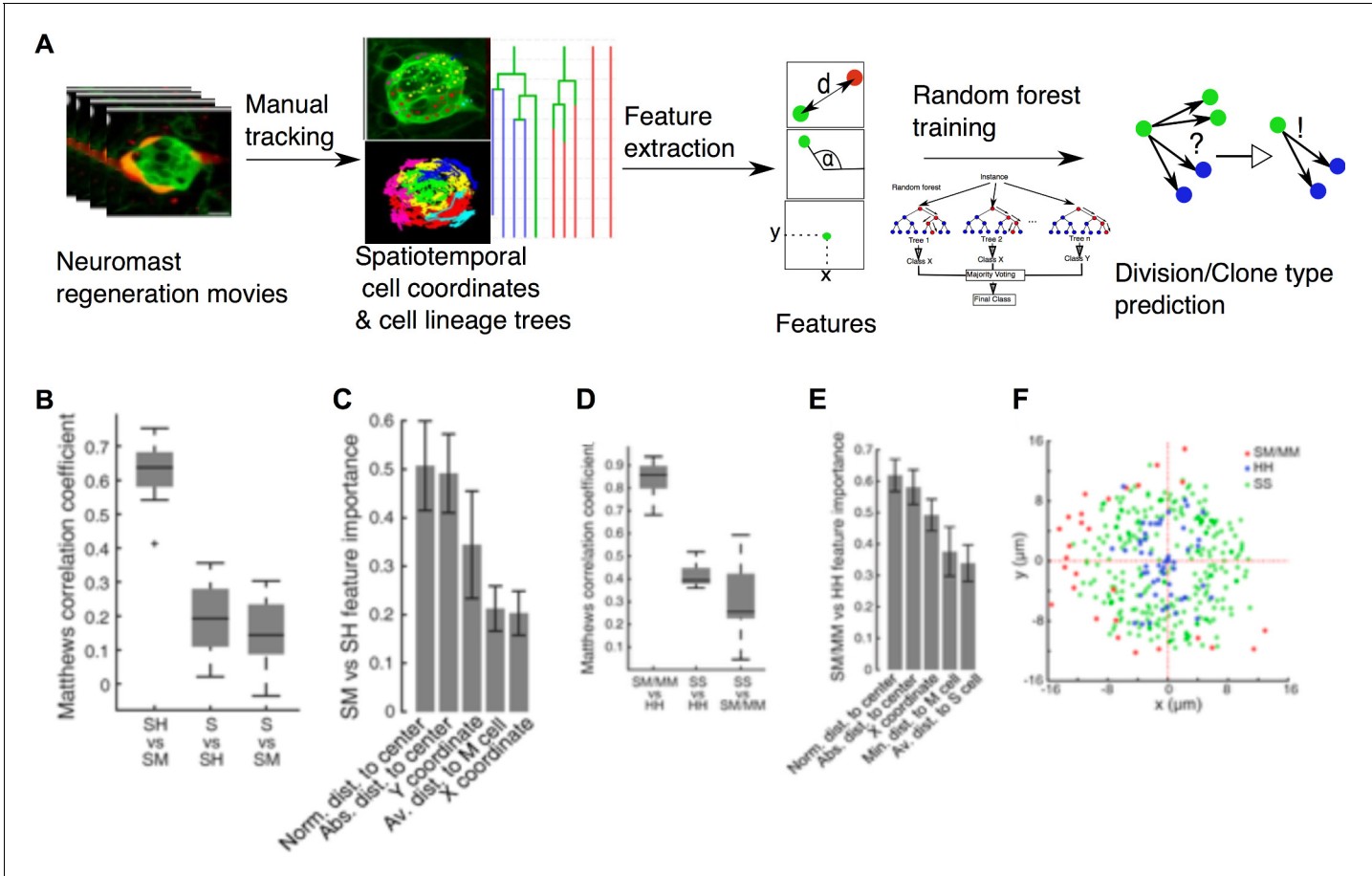

**Figure 8.** Implementation of predictive machine-learning analysis. (A) Overview from experiments to prediction. Movies of neuromast regeneration allow us to track every single cell over 100hpi and to generate a cell lineage from these track points. Information covered in all tracks and lineages can be extracted as features with which we train our random forest machine-learning classifier to predict division or cell lineage fate. (B) Sustentacular founder cell choices between SH vs. SM clones can be predicted with high accuracy (MCC = 0.63 ± 0.09, mean ± s.d., n = 15 bootstrapped samples) whilst choices between S and SH or SM clones are highly inaccurate (MCC = 0.19 ± 0.11 and 0.15 ± 0.10, mean ± s.d., respectively, n = 15 bootstrapped samples), based on 32 calculated features. (C) Features relative to the position of the founder cells and their nearest cellular environment can discriminate between SM and SH clone types. (D) Choices between SM/MM and HH divisions can be predicted with high accuracy (MCC = 0.91 ± 0.07, mean ± s.d., n = 15 bootstrapped samples) while those between SS and HH or SM/MM have low accuracy (MCC = 0.50 ± 0.05 and 0.38 ± 0.15, respectively, mean ± s.d., n = 15 bootstrapped samples) (E) Features describing the cell's position in relation to the neuromast center and their proximity to other mantle cells have the highest influence on the cell fate choices of a sustentacular cell. (F) SM/MM divisions (red) appear predominantly at the periphery of the organ whereas HH divisions (blue) appear proximal to the center. Sustentacular cell self-renewing divisions (SS, green) occur mostly around the neuromast center, generating a ring-like pattern.

DOI: https://doi.org/10.7554/eLife.30823.013

The following figure supplements are available for figure 8:

**Figure supplement 1.** Comparison of different classification methods.

DOI: https://doi.org/10.7554/eLife.30823.014

**Figure supplement 2.** Features used to predict SM vs SH clones sorted by predictive importance.

DOI: https://doi.org/10.7554/eLife.30823.015

**Figure supplement 3.** All features used to predict SM/MM vs HH divisions sorted by predictive importance.

DOI: https://doi.org/10.7554/eLife.30823.016

mantle cells respond to damage and contribute to the regenerative processes, and may drive the regeneration of an entire organ if every other cell class is lost.

One outstanding question is how regeneration is controlled spatially. The epithelium may respond to damage via dynamic formation of an injured-intact axis at the onset of repair. Our results support this scenario by unveiling the adaptability of the neuromast epithelium to the localization

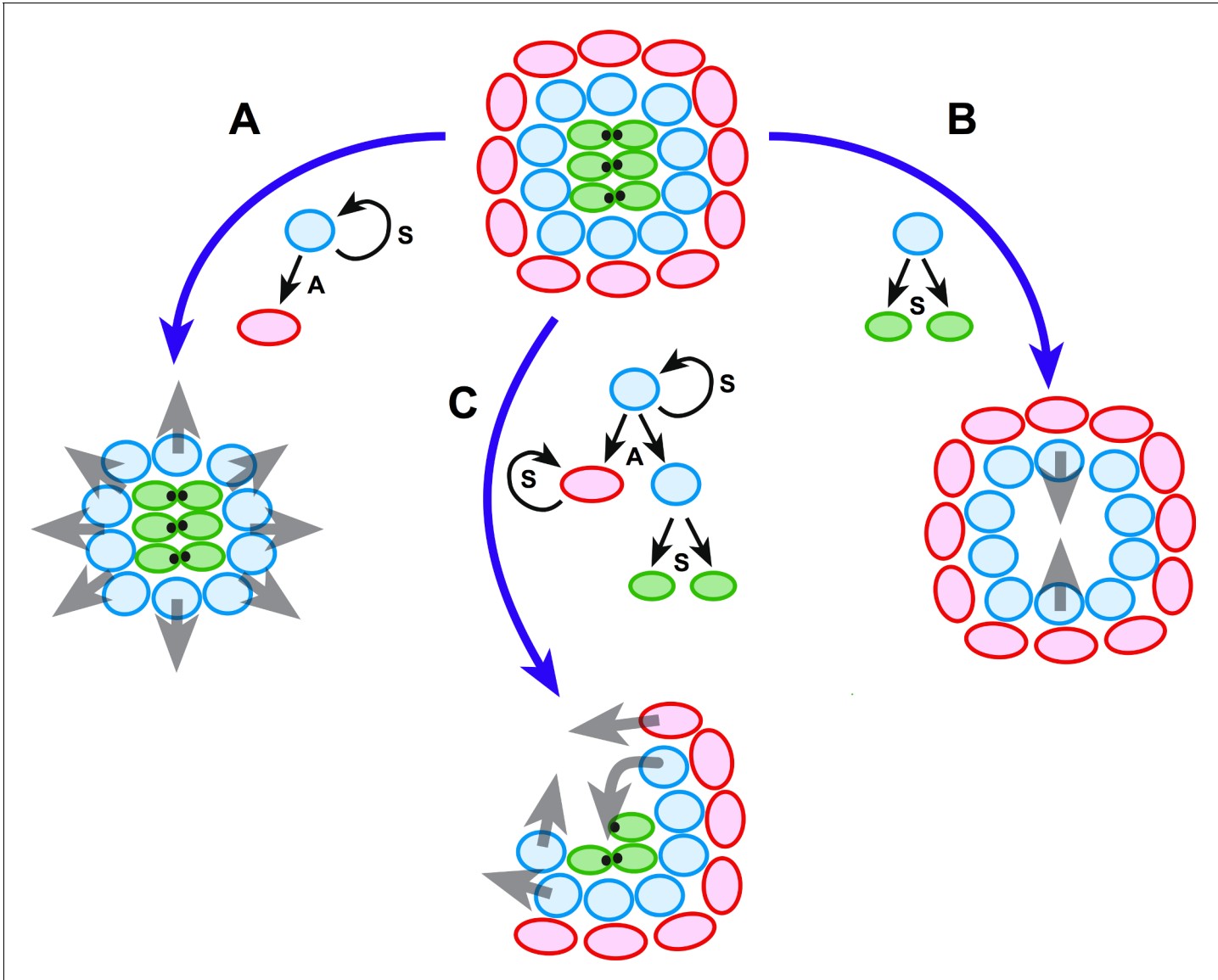

**Figure 9.** Schematic model of neuromast regeneration. The top diagram exemplifies the architecture of an intact neuromast. A, B and C indicate three types of injury: A when mantle cells are lost, B when hair cells are ablated, and C when a localized combination of all three cell classes is lost. Under the model that we present, radial symmetry serves to localize damage and canalize regeneration spatially. If central hair cells are lost (**A**), radial symmetry is maintained for sustentacular progenitors to regenerate hair cells centripetally (grey arrows in A). If outer cells are lost (**B**), radial symmetry is also maintained for the generation of progeny that will acquire mantle fate and propagate centrifugally to reform the outer rim of the neuromast (grey arrows in B). Upon asymmetric damage, however, the radial symmetry is partially broken (**C**). The neuroepithelium repolarizes along an injured-intact axis, which canalizes regeneration towards the damaged areas (grey arrows in C). Individual cells are color-coded (mantle cells in red, sustentacular cells in light blue, and hair cells in green), and in each case we indicate the type of division that the intact cells undergo: symmetric (**S**) when they produce two equivalent cells or self-renew, and asymmetric (**A**) when their division generates sibling cells that differentiate into different classes.
DOI: https://doi.org/10.7554/eLife.30823.017

and scale of damage. We suggest a model in which the invariant radial symmetry of the neuromast serves as a rheostat to identify the site of damage to guide regeneration spatially (*Figure 9*). A polarized axis along structurally intact and injured areas underlies this process. However, the complete reconstruction of a neuromast by as few as 4 cells suggests that a partial maintenance of radial symmetry is not essential for organ regeneration. Therefore, radial-symmetry maintenance cannot have a deterministic impact on the recovery of geometric order. Yet, partial structural maintenance and polarized tissue responses may optimize repair, respectively, by preventing superfluous cellular

production in undamaged areas and by biasing the production of lost cells in the damaged areas. For organs that have evolved under the pressure of persistent damage, compliance to the extent of the injury may be an advantage because the regenerative responses can be scalable and localized, allowing faster and more economical regeneration.

After the severest of injuries, regenerated neuromasts were plane polarized in a manner indistinguishable from unperturbed organs. This startling result indicates that as few as four founder supporting cells can re-organize the local coherent planar polarity of the epithelium during neuromast repair. An alternative explanation is that founder cells have access to external polarizing cues. One source of this information is an isotropic mechanical forces exerted by the interneuromast cells that flank a neuromast. This is possible because interneuromast cells are always aligned to the neuromast's axis of planar polarity. Yet, the concurrent ablation of resident hair cells and the interneuromast cells around an identified neuromast led to regenerated hair cells whose local orientation was coherent. Interestingly, recent studies have identified a transcription factor called Emx2 that regulates the orientation of hair cells in neuromasts of the zebrafish (*Jiang et al., 2017*). Emx2 is expressed in one half of the hair cells of the neuromast (those oriented towards the tail) and absent in the other half (which are coherently oriented towards the head). Loss- and gain-of-function of Emx2 alter planar cell polarity in a predictable manner: loss of Emx2 leads to neuromasts with every hair cells pointing towards the head of the animal, and Emx2 broad expression orients hair cells towards its tail. Because the coherent local axis of polarity is not affected by these genetic perturbations, Emx2 may act in hair cells as a decoder of global polarity cues. This evidence, together with our results, suggests that during neuromast regeneration founder cells autonomously organize the variegated expression of Emx2 in the regrowing epithelium with consequent recovery of a coherent axis of planar polarity and with one half of the hair cells pointing opposite to the other half. The future development of live markers of Emx2 expression will be able to test this prediction. We would like to highlight that we do not currently understand the global polarization of the neuromast epithelium relative to the main body axes of the animal. External sources of polarity may impinge in the recovery of these global axes during neuromast regeneration. Previous work has demonstrated that local and global polarization occur independently of innervation (*López-Schier and Hudspeth, 2006*), but other potential polarizing cues remain untested. Therefore, at present we can only support the notion that local coherent polarity is self-organizing, whereas global orientation may be controlled externally.

Our results beg the question of whether neuromast cells self-organize. Our operational definition of self-organization is an 'autonomous increase in order by the sole interaction of the elements of the system' (*Haken, 1983*), implying that a cellular collective organizes a complex structure without the influence of external morphogenetic landmarks, patterning cues, or pre-existent differential gene-expression profiles. If these conditions are not met, cellular groups may nevertheless form a complex structure through a process of 'self-assembly' (*Sasai, 2013*; *Turner et al., 2016*). The reduction of neuromasts to around 5% of their original size shows that intact resident cells can rapidly recreate their original microenvironment to rebuild a neuromast with normal organization, proportions and polarity. Although these observations suggest autonomy, extrinsic sources of information including the extracellular matrix that remains intact after cell loss may serve as a blueprint for epithelial organization. Yet, unless such patterns are rebuilt together with the organ, neuromasts architecture and proportions would depend on the area occupied by the regrowing epithelium. In other words, cell-fate acquisition and cell-class distribution must be tissue-size dependent. However, we show that neuromast regain geometric order as early as 2 days after injury, when their cellular content is less than 60% of the original. Although our results do not irrefutably demonstrate self-organization during neuromast regeneration, they strongly support this idea. We argue that self-organization is an optimal morphogenetic process to govern organ repair because (i) it requires the least amount of previous information and (ii) it is robust to run-off signals that could lead to catastrophic failure.

## Conclusions

Understanding how tissues respond to the inherently random nature of injury to recapitulate their architecture requires the identification of cues and signals that determine cell-fate acquisition, localization and three-dimensional organization. Here we reveal an archetypal sensory organ endowed with isotropic regenerative ability and responses that comply to damage severity, nature and localization. An important corollary of our findings is that progenitor behavior is a facultative status that every

**Table 1.** List of prediction features with description.

We used 32 mainly spatial and neighborhood specific features for the classification. Features are explained in the description column.

| Feature name | Description |
| --- | --- |
| Absolut time | Hours post induction (hpi) |
| Absolute distance to center | Euclidean distance to the neuromast center |
| Average distance to H cell | - |
| Average distance to M cell | - |
| Average distance to S cell | - |
| Cell generation | Number of divisions that the cell has undergone |
| Founder Cell Type | - |
| Minimum distance to H cell | - |
| Minimum distance to M cell | – |
| Minimum distance to S cell | - |
| Movement angle to last division | Angle between current cell location, neuromast center and location of last cell division (or start of the movie in case of founder cell division) |
| Movement direction compared to center | Radial distance between current cell location and location of last cell division (or start of the movie in case of founder cell division). If the current location is nearer to the center the value is (+) in case it is further away the value is (-) |
| Movement distance since last division | Euclidean distance between current cell location and last cell division (or start of the movie in case of founder cell division) |
| Normalized distance to center | Radial distance of current cell location divided by the radial distance of the current furthest cell (to approximate the neuromast size) |
| Number of founder cells | - |
| Number of H cells | - |
| Number of H cells in 10 μm radius | - |
| Number of H cells in 20 μm radius | - |
| Number of H cells in 30 μm radius | - |
| Number of M cells | - |
| Number of M cells in 10 μm radius | - |
| Number of M cells in 20 μm radius | - |
| Number of M cells in 30 μm radius | - |
| Number of S cells | - |
| Number of S cells in 10 μm radius | - |
| Number of S cells in 20 μm radius | - |
| Number of S cells in 30 μm radius | - |
| Number of total cells | - |
| Polar angle | Polar angle is the counterclockwise angle between the x-axis, the neuromast center and the current cell location |
| Time to last division | Time to last division (or start of the movie in case of founder cell division) |
| X coordinate | - |
| Y coordinate | - |

DOI: https://doi.org/10.7554/eLife.30823.012

sustentacular cell can acquire or abandon during regeneration (*Blanpain and Fuchs, 2014*; *Wymeersch et al., 2016*). Importantly, we illustrate a machine learning implementation to identify features that predict cell-fate choices during tissue growth and morphogenesis. This quantitative approach is simple and model-independent, which facilitates its application to other organs or experimental systems to understand how multiple cells interact dynamically during organogenesis and organ regeneration in the natural context of the whole animal, and to identify how divergences from the normal regenerative processes lead to failed tissue repair.

## Materials and methods

### Zebrafish strains and husbandry

Zebrafish were maintained under standard conditions, and experiments were performed in accordance with protocols approved by the PRBB Ethical Committee of Animal Experimentation of the PRBB Barcelona, Spain. Eggs were collected from natural spawning and maintained at 28.5°C in Petri dishes at a density of up to 50 per dish. Transgenic lines used were *ET(krt4:EGFP)SqGw57A* (referred to in the text as SqGw57A) (*Kondrychyn et al., 2011*), *ET(krt4:EGFP)SqET4* (SqET4) (*Parinov et al., 2004*), *Tg[Myo6b:actb1-EGFP]* (*Kindt et al., 2012*), *Tg[−8.0cldnb:Lyn-EGFP]* (Cldnb:lynGFP) (*Haas and Gilmour, 2006*), *Tg[Alpl:mCherry]* (*Steiner et al., 2014*), *Tg[Sox2-2a-sfGFPstl84]* (referred to as Sox2:GFP) (*Shin et al., 2014*). To label cell nuclei, in vitro transcribed capped RNA coding for histone 2B-mCherry was injected in 1–4 cell embryos at a concentration of 100 ng/µl (*Rosen et al., 2009*). Throughout the study, zebrafish larvæ were anesthetized with a 610 µM solution of the anesthetic 3-aminobenzoic acid ethyl ester (MS-222).

### Laser-mediated cell ablations

For *in toto* cell ablation, we used the iLasPulse laser system (Roper Scientific SAS, Evry, France) mounted on a Zeiss Axio Observer inverted microscope equipped with a 63X water-immersion objective (N.A. = 1.2) (*Xiao et al., 2015*). The same ablation protocol was used for all experiments using five dpf larvæ. Briefly, zebrafish larvæ were anesthetized, mounted on a glass-bottom dish and embedded in 1% low-melting point agarose. Three laser pulses (355 nm, 400 ps/2.5 µJ per pulse) were applied to each target cell. After beam delivery, larvæ were removed from the agarose and placed in anesthesia-free embryo medium. All ablations were systematically performed on the L2 or L3 posterior lateral-line neuromasts, except for those in *Figure 6F*, for which we targeted the LII.2 neuromast.

### Phalloidin staining

Samples were fixed in 4% PFA overnight at 4°C, washed several times in 0.1% PBSTw and incubated in phalloidin-Alexa 568 or Alexa 488 (Invitrogen) diluted 1:20 in 0.1% PBSTw overnight at 4°C. Samples were washed several times in 0.1% PBSTw and mounted in 0.1% PBSTw with Vectashield (1/100, Vector Labs, Burlingame, CA, USA).

### Regeneration analysis and quantification

For quantification of cell numbers during neuromast regeneration, *Tg[Cldnb:lynGFP; SqGw57A; Alpl:mCherry]* zebrafish larvæ were anesthetized, mounted on a glass-bottom dish and embedded in 1% low-melting point agarose. All samples were imaged before injury, 4 hpi and every 24 hr up to 7 dpi with an inverted spinning-disc confocal microscope (Zeiss by Visitron), under a 63X water-immersion objective. After imaging, larvæ were quickly transferred to anesthetic-free medium. Cells were manually counted using the FIJI multi-point tool by scrolling throughout the entire volume of the neuromast. Cell classes were identified by the following criteria: Interneuromast cells: Cldnb:lynGFP(+), SqGw57A(-), Alpl:mCherry(+). Mantle cells: Cldnb:lynGFP(+), SqGw57A(+), Alpl:mCherry(+). Sustentacular cells: Cldnb:lynGFP(+), SqGw57A(+), Alpl:mCherry(-). Hair cells: Cldnb:lynGFP(+), SqGw57A(-), Alpl:mCherry(-). Hair cell identity was verified by the concomitant observation of the correct transgene expression pattern, central-apical location and the presence of a hair-cell bundle. Data was processed and analyzed using GraphPad Prism version 6.04 for Windows (GraphPad Software, La Jolla, CA, USA, www.graphpad.com). In the box plots, the boundary of the box closest to zero indicates the 25th percentile (q1), a black line within the box marks the median, and the boundary of the

box farthest from zero indicates the 75th percentile (q3). Whiskers above and below the box include points that are not outliers. Points are considered as outliers if they are bigger than q3 + 1.5(q3 – q1) or smaller than q1 – 1.5(q3 – q1).

## Videomicroscopy, cell tracking and lineage tracing

Larvæ were anesthetized, mounted onto a glass-bottom 3 cm Petri dish (MatTek) and covered with 1% low-melting point agarose with diluted anesthetic. Z-stack series were acquired every 15 min at 28.5°C using a 63X water-immersion objective. Cells were tracked overtime using volumetric Z-stack images with FIJI plugin MTrackJ (*Meijering et al., 2012*). Movies were registered two times for image stabilization and centered upon the centroid of the surviving group of cells and the subsequent regenerating organs. Founder cells are identified from 1 to 6 (*n*) and their daughter cells receive *2* n and *2n + 1* identities. All images were processed with the FIJI software package.

## Pharmacology

All pharmacological treatments were performed as described previously (*López-Schier and Hudspeth, 2006*; *Wibowo et al., 2011*; *Pinto-Teixeira et al., 2015*). Briefly, the following concentrations and timings used were: Neomycin sulfate (Sigma, St. Louis, MO) 250 µM for 45 min; N-[N-(3,5-difluorophenacetyl)-L-alanyl]-S-phenylglycine-t-butyl ester (DAPT) (Sigma) 100 µM for 24–48 hr. Equal amounts of DMSO were diluted in embryo medium for control specimens.

## Random forest prediction

Random forest algorithms use the majority vote of numerous decision trees based on selected features to predict choices between given outcomes (*Murphy, 2012*). We used a list of spatial, movement and neighborhood features (see Suppl. *Table 1*) to perform the random forest prediction of fate choice. We trained the random forest on 14 experiments and tested our prediction on one left-out experiment in a round robin fashion, leading to 15 test sets overall. To evaluate our prediction, we calculated Matthews correlation coefficient (MCC) (*Matthews, 1975*), which accounts for imbalance in our data (e.g. 78% of all divisions are SS divisions). The MCC is calculated by:

$$\text{MCC} = \frac{TP \times TN - FP \times FN}{\sqrt{(TP + FP)(TP + FN)(TN + FP)(TN + FN)}}$$

where TP denotes true positive, TN true negative, FP true positive and FN false negative predictions. The MCC can have values between −1 and +1, where −1 is a completely incorrect, 0 a random and +1 a perfect prediction. To evaluate the variance of the MCC on the 15 test sets we used a bootstrapping approach, where we draw 15 samples from all test sets with replacement 15 times. From this resampled data we calculated the mean MCC and the standard deviation as shown in *Figure 8B and D*. All machine-learning analyses were performed using MATLAB (Version 2015b on a Windows 7 machine)

## Acknowledgements

We thank A Steiner, T Nicolson and L Solnica-Krezel for transgenic zebrafish, the animal facility personnel at the CRG of Barcelona and the HMGU for animal care, and Kirill Smirnov for statistical support. Funding was provided by the European Research Council Grant 2007_205095, by the ESF Research Networking Programme 'QuanTissue', and by the AGAUR Grant 2009-SGR-305 of Spain.

## Additional information

### Funding

| Funder | Grant reference number | Author |
| --- | --- | --- |
| European Research Council | 2007_205095 | Hernán López-Schier |
| Agència de Gestió d'Ajuts Universitaris i de Recerca | 2009-SGR-305 | Hernán López-Schier |

The funders had no role in study design, data collection and interpretation, or the decision to submit the work for publication.

## Author contributions
Oriol Viader-Llargués, Data curation, Formal analysis, Validation, Investigation, Visualization, Methodology, Writing—original draft, Writing—review and editing; Valerio Lupperger, Resources, Software, Methodology, Writing—review and editing; Laura Pola-Morell, Resources, Investigation, Writing—review and editing; Carsten Marr, Software, Formal analysis, Supervision, Methodology, Writing—review and editing; Hernán López-Schier, Conceptualization, Supervision, Funding acquisition, Visualization, Writing—original draft, Project administration, Writing—review and editing

## Author ORCIDs
Hernán López-Schier ⓘ https://orcid.org/0000-0001-7925-7439

## Ethics
Animal experimentation: Zebrafish were maintained under standard conditions. Experiments with wild-type, mutant and transgenic embryos of undetermined sex were conducted in accordance with institutional guidelines and under a protocol approved by the Ethical Committee of Animal Experimentation of the Parc de Recerca Biomedica de Barcelona, Spain, and protocol number Gz.:55.2-1-54-2532-202-2014 by the "Regierung von Oberbayern", Germany.

## Decision letter and Author response
Decision letter https://doi.org/10.7554/eLife.30823.020
Author response https://doi.org/10.7554/eLife.30823.021

# Additional files

## Supplementary files
• Transparent reporting form
DOI: https://doi.org/10.7554/eLife.30823.018

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
