## [Decision Letter]

[Editors’ note: this article was originally rejected after discussions between the reviewers, but the authors were invited to resubmit after an appeal against the decision.]

Thank you for submitting your work entitled "Multicellular self-organization underlies the accuracy of sensory-organ regeneration" for consideration by *eLife*. Your article has been reviewed by three peer reviewers, and the evaluation has been overseen by Tanya Whitfield as Reviewing Editor and a Senior Editor. The reviewers have opted to remain anonymous.

Our decision has been reached after consultation between the reviewers. Based on these discussions and the individual reviews below, we regret to inform you that your work will not be considered further for publication in *eLife*.

There was praise among the reviewers for the rigorous and careful quantitative analysis of cell behaviour during neuromast regeneration. However, the overall consensus was that the manuscript tackled many disparate aspects, without focussing in depth on any one of them; it therefore lacked a clear 'big message'. In their discussions, the reviewers highlighted the lack of a significant conceptual advance, together with missing citation or discussion of some of the previous literature that would help to explain or support the results. These included Jiang, Kindt and Wu; 2017; Kempfle et al., Sci Rep, 2016 and Seleit et al., Development, 2017.

Suggestions for improving the manuscript included a cut of earlier sections of the manuscript and a deeper analysis of the later ones; or, placing the results in the context of known molecular pathways of neuromast regeneration. The full reviews, which are appended below, have a number of more specific concerns and suggestions.

*Reviewer #1:*

The authors of this study proposed a "self-regulatory mechanism that guides the regenerative process (of neuromasts) to identical outcome despite the intrinsically stochastic nature of damage".

While their data largely support the authors’ conclusion, their analysis of potential mechanisms and discussion of how this study enriches what is already known about hair-cell organ regeneration are lacking. Specifically, there were a number of points in the Results and Discussion where conclusions were stated, yet references to recent studies that would support those conclusions and provide potential mechanisms for what the authors observed were notably absent. In addition, there were a few experiments where I was unsure about the authors’ controls.

Point by point concerns are addressed below:

- Results: "laser targeted cells undergo rapid apoptosis" needs a citation or supporting data.

- Results: "ablated neuromasts and their neighboring Schwann cells without affecting the adjacent interneuromast cells"; both the interneuromast cells and Schwann cells contain the mcherry fluorophore, correct? How did the authors verify that the interneuromast cells were not damaged when nearby Schwann cells were ablated?

- Results: is there anything in the literature that would indicate that inhibition of ErB kinase would initiate production of additional neuromasts at 5 dpf i.e. following the migration and establishment of the posterior lateral-line organs?

- Results: The result observed after treating ablated neuromasts with AG1478 is interesting, but I'm puzzled as to why the authors didn't take it one logical step further and address whether wnt/bcatenin signaling is absent i.e. could they initiate interneuromast cell proliferation with exposure to BIO?

- Figure 3K-O: By eye, dorsal mantle cell recovery doesn't look very robust. Is this a representative example? The total cell numbers in Figure 3Z don't really address if certain classes of cells fully recover.

- Figure 4—figure supplement 1 and its related Discussion: I'm not quite sure what the authors claim regarding amount and timing of genetically encoded marker expression means. Are they using GFP reporter expression as a proxy for actual gene expression? Is so, it needs to be supported by endogenous expression data (e.g. insitus) or references to previous studies that demonstrate exogenous reporter expression reflects endogenous gene expression in these reporter lines.

- Results: When describing the combination of reporter lines used, it would be helpful to the reader to put the name of the line next to the cellular structures they label in the text.

- Figure 4U-W; X-Z: By eye, the polarity of phalloidin labeling looks mixed, and X-Z is further confounded by the presence of immature hair-cells. To better make the case for fully reformed radial symmetry 1) a control, non-ablated NM should be shown for comparison 2) the axis of symmetry should be indicated with a hashed line 3) relative orientation of HC per NM should be quantified.

- The observation that a few regenerating hair cells recovered planar polarity could be explained by Emx2 expression in 1 of 2 sibling hair cells (Jiang, Kindt and Wu, 2017). It's important to discuss this potential mechanism either here or in the Discussion section even if the authors didn't test for Emx2 expression in this study.

- Subsection “Sustentacular and mantle cells have different regenerative potential”, the conclusion that *Sox2*(+) sustentacular cells are tripotent is supported by evidence from a previous study; *Sox2* has been shown to both expand progenitor cells and initiate their differentiation in the cochlea (Kemfle et al., Sci Rep, March 2016).

- In the same section, "predecing". Typo.

- "Assessed spatial distribution…60 hrs of regeneration" which ablation protocol did they use here?

- Figure 5E: Both supporting cells and hair cell are labeled with GFP. What criteria did the authors use to differentiate supporting cells from hair cells?

- Figure 6 K-L: Cell cycle length appears strongly correlated to regeneration time (K), but not so strongly correlated to neuromast size overall. The legend and the text states in L that the cell cycle length doesn't correlate well until 24 cells, but that doesn't seem directly apparent from the graph or the regression line.

- Subsection “*Sox2*(+) sustentacular cells are equipotent and plastic”: There is no mention of Figure 7C in the results, and it's unclear to me from the figure legend how this data was generated.

- Discussion section; "suggesting that *Sox2* itself is not essential" is overreaching. The observation made from interneuromast cells may show that *Sox2* is not sufficient to drive regeneration, but it does not demonstrate it's not essential.

- Discussion section, second paragraph: Not mentioning the Emx2 study here is a glaring omission.

- The Discussion overall would benefit from a more thorough incorporation of molecular mechanisms reported in previous studies examining supporting cell proliferation, differentiation, and repolarization in hair-cell organs to understand how this study's results enriches our understanding of the self-regulatory processes that guide lateral-line organ restoration.

*Reviewer #2:*

This study focus on one of the most intriguing and fascinating aspects or regenerative biology: the accuracy with which organs respond to injuries to restore proper size and cellular organization. The authors use neuromasts of fish to explore this phenomenon and fully exploit the advantages of the system. The videos and figures are of very high quality and aesthetic value, and the authors extract from them an exhaustive quantification of cellular responses during regeneration. I find the manuscript nicely written and in most sections easy to follow.

I failed, however, to understand what is the main new finding that the authors are presenting. It is known from several models that tissues (epidermis), organs (liver, skin, the fish heart), limbs (axolotls and newts, caudal fin in fish), systems (hematopoietic) and even organisms (planarians) can re-generate accurately to their pre-injury state. The authors state clearly some of the open questions in the field, but it feels they fall short at addressing them thoroughly when using the neuromast. The major achievement of the manuscript is the acquisition of an extensive dataset on individual cellular behavior during the regeneration response. The two main points I see are that neuromasts regenerate what is lost, and cell identity is acquired along a radial axis. Unfortunately and despite the extensive analysis performed and the quality of the data, it is hard to understand the conceptual novelty of this study.

Major comments:

During the Introduction and the initial section of Results, it feels that the authors neglect several findings from a previous publication of another group, Sánchez et al., 2016, on the regeneration of neuromasts upon ablation of the entire organ. The results presented in the submitted paper do not match a proportion of the results previously published (neuromast regeneration upon ablation or pharmacological inactivation of Schwann cells). The authors should explain this discrepancy and acknowledge that organ regeneration after severe injuries can indeed happen from other sources as well.

The authors state that inter neuromast cells are not necessary for the regeneration response. They show however that in their absence, organs regenerate to a smaller size. What is then the role of interneuromast cells?

In reference to the autonomous regeneration of epithelial polarity the authors show that neuromasts regenerate the proper polarisation after severe injuries in the presence of interneuromast cells, or after ablation of hair cells in the absence of interneuromast cells. To claim an autonomous process, they should look at the polarisation plane after severe injuries and in the absence of interneuromast cells – experiments done for a previous section but not analyzed for epithelial polarity. In addition, the authors should consider that anisotropic forces might originate not only from interneuromast cells and therefore evaluate whether this is the valid conclusion from their experimental data.

The authors emphasize the accuracy of the regenerative response, but this seems to be restricted to the distribution of cell types. The number of cells, and therefore the size of the organ, is restored only to a 70% after 7dpi. This needs to be stressed, taken in consideration for the conclusions, and contemplated in the Discussion.

It is still not clear whether neuromast cells acquire a proper position during regeneration or whether they acquire their fate based on the position they occupy. The authors state both claims along the manuscript although they are mutually exclusive.

The claim that sustentacular cells revert to an embryonic (undifferentiated) state immediately after tissue injury is based in the transient down-regulation of a single fluorescent reporter. The authors should present data for some of the many additional markers or tone down their conclusions. On a more general view, I disagree with the claim that undifferentiated equals embryonic.

*Reviewer #3:*

The work by Oriol Viader-Llargués et al. examines how zebrafish mechanosensory organs, neuromasts (NMs), respond to injury. The authors ablate portions of various sizes from different regions of the organ and then examine organ regeneration using live imaging of transgenes that mark distinct cell lineages within NMs. A number of previous studies examined regeneration of a single NM cell type, hair cells; however, regeneration capacity of other cell types has not been studied. While carefully done, the study is largely descriptive and does not build on the previous knowledge of molecular mechanisms that drive NM development and regeneration. Nevertheless, this work presents a number of novel findings, related to the cellular mechanisms of organ regeneration that should contribute significantly to the field of organ regeneration: 1) NMs can fully regenerate up 70 to 90% of their original size regardless of injury site; 2) regeneration process restores relative proportion and proper polarity of NM cell types; 3) regeneration is based on a resident population of *Sox2*-positive cells (not on stem cells) that restore all cell types within NMs; 4) even severely damaged organs (as few as 4 cells left) can fully regenerate; 5) cell fate during regeneration can be predicted based on the distance of dividing progenitors from the center. I have a few specific comments related to data interpretation and conclusions.

As the authors point out, previous studies looking at hair cell regeneration found that the source of hair cell progenitors resides in NM poles. Present study found that *Sox2*^+^ progenitors throughout the NM contribute to hair cells. How do they reconcile the differences? This should be discussed.

Ablation of inter-NM cells led to smaller regenerating NM after injury. Again, this issue is not discussed.

The authors conclude that *Sox2*^+^ cells are resident progenitors that potentially dedifferentiate following injury and then contribute to all three distinct cells types in the NM. Have they tried eliminating *Sox2*^+^ cells to support this assertion?

Related to the last point, it would certainly add to the impact of the paper if the authors can show that *Sox2*^+^ cells de-differentiate, based on molecular markers.

[Editors’ note: what now follows is the decision letter after the authors submitted for further consideration.]

Thank you for resubmitting your work entitled "Live cell-phylogeny tracing and machine learning reveal patterns of organ regeneration" for further consideration at *eLife*. Your revised article has been favorably evaluated by Naama Barkai as Senior editor, Tanya Whitfield as Reviewing editor, and three reviewers.

The three reviewers are all very positive about the revisions to the manuscript, and appreciate the care that has gone into the revisions. However, all three have some additional comments and queries that should be addressed. These mostly involve requests for further discussion and context, together with correction of typos, and should be quick to complete. The full reviews are appended here for your information.

*Reviewer #1:*

The revised version of the manuscript by Oriol Viader-Llargués et al. is significantly improved from the previous version. Its message is more focused, its conclusions better supported by the data, and references to relevant work were incorporated.

With that said, I have a few major questions that don't necessarily need to be addressed with additional experiments:

1) The mantle cell linage data show that the mantle cells do not give rise to any cell class other than their own, leading to the conclusion that mantle cells are not essential contributors to neuromast repair. Yet none of their injury paradigms leave only the mantle cells intact i.e. every injury with mantle cells present also has sustentacular cells present. It could be the case that mantle cells are essential to regenerating a neuromast if the sustentacular cells are absent.

The authors refer to a study in their rebuttal that suggests mantle cells are capable of producing all neuromast cell classes in a regenerating fin. Do the authors predict this would be the case, if they ablated everything but the mantle cells? If not, why?

2) The authors discussed the potential influence of interneuromast cells on planar polarity, but did not elaborate influence of INCs on neuromast size (Figure 4K-P). The result is striking and should be discussed further than the statement in the Results-"non-essential, yet appreciable contribution to regeneration".

*Reviewer #2:*

In the new version of this manuscript, Viader-Llargues, Lopez-Schier, and colleagues present their results with a more clear aim, focusing on quantitative aspects of the regeneration response and new tools to investigate it. My main points were properly addressed, namely toning down certain conclusions, including alternative scenarios and focusing on defined topics rather than superficially following many. I appreciate particularly the scientific quality and dedication of the authors' replies to many of my previous concerns. The lack of overshooting of cell types during the regeneration response was a very nice addition that contrasts to other regenerative systems, and illustrates how controlled the regenerative response is in neuromasts.

These are my comments on the current version:

In the title and along the manuscript the authors use the term "cell-phylogeny tracing". Is there any reason not to use the widely accepted term "lineage" instead? I feel that "lineage" works better in this context since it reflects more accurately the continuum of the data that the authors have acquired.

The issue of self-regulation of the regenerative response is dealt more carefully than in the previous version. Still, I would like the authors to be more explicit about their interpretation of the data, mainly on the Discussion. One of the aspects the authors focus on is the re-establishment of a polarity axis, and this is a very interesting aspect of the regeneration response since polarity is heterogeneous among different neuromasts of the posterior lateral line – unlike cell types and their distribution. In other words: neuromasts have one solution for the cell-type problem, but two solutions for the polarity problem. Previous work from the group (Lopez-Schier et al., 2004 and Lopez-Schier and Hudspeth, 2006) has clearly stated that the polarity axis is related to the final migration of the neuromasts during organogenesis. I find troubles imagining how a self-renewing organ (or 4-10 cells, to put it bluntly) will always choose the original polarity in the absence of external cues. I agree with the authors that external cues have minor roles in the establishment of cells types or position of cells, but extending self-organization to the re-establishment of polarity seems inappropriate to me. Am I missing something? Maybe the authors could speculate about the role of the afferent axons (I guess they should remain to some extent under the injury paradigms used in this study), which they have shown to display a high accuracy recognizing hair cells of a given polarity (Pujol-Marti, Current Biology 2014).

I particularly enjoyed the reply of the authors regarding the differences and similarities in mantle cell behavior during homeostasis and regeneration. I think it would be an added value if they incorporate these concepts in the Discussion.

Results section, paragraph two; I believe that the authors should include a reference to Grant et al., 2005. In fact, the authors do so in their response to Reviewer #1.

Subsection “Sustentacular cells are equipotent and plastic”. I found this part difficult to understand. Either the written numbers do not match what the authors show between brackets, or there is some mistake in the annotation. What is "peak at 8h, mean+-s.d at 14+-9 hours"? Overall, I find this part the less clearly written of the manuscript. Also, is it valid to call a peak sharp with such a big s.d.?

Discussion paragraph one: The authors state that sustentacular cells are equipotent, which I think is an overstating.

The data presented in the manuscript reveals that 4 or more of them are enough to regenerate the entire organ. The use of equipotency, in my view, states that all of them do the same during the process. In their accurate data in Figure 7, they show that some of them generate M cells, some other S cells, and some other combinations of H and other cell types. I understand that equipotent members of a population could behave in different manners based on either stochastic internal programs or external cues, but to prove which is the case demands a more extensive dataset and a deep mathematic analysis, which is not the scope of the present study – although this is a fantastic system to tackle it! I feel that the use of equipotent in this context assumes features that were not tested experimentally. I suggest the authors stay with "tri-potent", a term that they have used along Results.

*Reviewer #3:*

This is a revised version of a manuscript by Oriol Viader-Llargués et al. that deals with cellular mechanisms of neuromast regeneration. The previous version was criticized by its descriptive nature, lack of focus and omission of some references. The authors largely remedied these issues, although I still feel that some of their findings are not sufficiently discussed in the context of previous studies.

Comments:

The authors mention that their findings contrast those of Sanchez et al. study, but do now offer any discussion as to why that is the case. I think it is important to offer at least some explanation.

It seem that examples of severe ablation following full regeneration always leave at least one mantle cell (“Neuromast architecture recovers after severe loss of tissue integrity”). If this is indeed the case, it may indicate that a combinatorial signal(s) is required to initiate organ regeneration.

There are only two cases when sustentacular cell progenitors gave rise to all three cell types. Based on these small numbers, I think the authors need to be careful concluding that a sustentacular cell is a multipotent progenitor for all cell types in the neuromast.

The finding that at least 4 cells are needed to reconstitute an organ is intriguing. This is reminiscent of planarian organ regeneration where 1/300th part but not less can reconstitute full animal. It was later discovered that this is a minimum fraction of the animal roughly containing at least one stem cell (neoblast) necessary to regenerate all tissue lineages. Again, some discussion, as to why the authors think this is case (4 cells but not fewer are required) is warranted

Discussion paragraph two, final sentence: Seleit et al., 2017 showed that "new cell type" exists in zebrafish. They also showed that mantle cell are neuromasts stem cells. Thus, it is important to discuss how homeostatic cell renewal differs from cell renewal during NM regeneration (i.e. mantle cell as stem cell during homeostasis vs. sustentacular cell as a multipotent progenitor during regeneration). This is an interesting question, as there are examples where cells can be driven to change their fate by extreme injury paradigm.

---

## [Author Response]

[Editors’ note: the author responses to the first round of peer review follow.]

Reviewer #1:The authors of this study proposed a "self-regulatory mechanism that guides the regenerative process (of neuromasts) to identical outcome despite the intrinsically stochastic nature of damage".While their data largely support the authors’ conclusion, their analysis of potential mechanisms and discussion of how this study enriches what is already known about hair-cell organ regeneration are lacking. Specifically, there were a number of points in the Results and Discussion where conclusions were stated, yet references to recent studies that would support those conclusions and provide potential mechanisms for what the authors observed were notably absent. In addition, there were a few experiments where I was unsure about the authors’ controls.

We thank the reviewer for these comments, who indicates that some data were sub-optimally discussed. We believe that this led to the critique that the work lacks novelty. Consequently, we (i) have thoroughly revised the discussion and the references to include important previous work, (ii) emphasize the novel aspects of our work, and how our conclusions could not have been predicted by previous results, and (iii) have reduced parts of the paper that were not central to the main message, and expanded others by including control experiments and further quantitative analyses. Specifically, we include new panels in Figure 1 that reveal the accuracy and specificity of our laser-mediated cell-ablation protocol, and an accompanying supplemental video. We have also rearranged Figures 6 and 7 and generated a new figure to explain the machine learning approach.

Point by point concerns are addressed below:- Results: "laser targeted cells undergo rapid apoptosis" needs a citation or supporting data.

We thank the reviewer for raising this point. Indeed, we do not demonstrate that laser ablation triggers apoptotic cell death. Thus, we now state that: “we certified that laser-targeted cells are rapidly eliminated from the neuromast epithelium with no detectable collateral damage”. We include new data showing target-cell elimination in Figure 1K-P and Video 1.

- Results: "ablated neuromasts and their neighboring Schwann cells without affecting the adjacent interneuromast cells"; both the interneuromast cells and Schwann cells contain the mcherry fluorophore, correct? How did the authors verify that the interneuromast cells were not damaged when nearby Schwann cells were ablated?

In neuromasts of the transgenic lines that we have used, interneuromast cells and Schwann cells express different fluorescent proteins, which can be easily distinguished. Precisely to be able to distinguish between these cell classes is the reason behind our generation of a quintuple transgenic line for these experiments. We understand that the size of the images may have prevented the reviewer to appreciate these differences. However, because of the refocus of the work on the technical approach, these data have become superfluous and were eliminated. Please, note that these changes better define the message of the paper without altering its original conclusion.

- Results: is there anything in the literature that would indicate that inhibition of ErB kinase would initiate production of additional neuromasts at 5 dpf i.e. following the migration and establishment of the posterior lateral-line organs?

Yes, there is, and we have cited the relevant publications for the pharmacological inhibition of ErbB (Sánchez et al., 2016), as well as the genetic ablation of the receptor controlling the signal (Grant et al., 2005; López-Schier and Hudspeth, 2005). Yet, as stated above, the new focus of the paper has made this part of the results superfluous.

- Results: The result observed after treating ablated neuromasts with AG1478 is interesting, but I'm puzzled as to why the authors didn't take it one logical step further and address whether wnt/bcatenin signaling is absent i.e. could they initiate interneuromast cell proliferation with exposure to BIO?

This is a valid critique and a very interesting question that has not escaped our attention. However, a thorough investigation of Wnt signaling within the context of organ-size control would result in a longer and less focused paper. Nevertheless, we have added a paragraph that discusses cell proliferation during neuromast formation. Specifically, we state in the Discussion that “we did not observe regenerative overshoot of any cell class (Agarwala et al., 2015), suggesting the existence of a mechanism that senses the total number of cells and the cell-class balance during tissue repair (Simon et al., 2009). Previous work indicates that such mechanims may be based on interplay between FGF, Notch and Wnt signaling (Ma et al., 2008; Wibowo et al., 2011; Wada et al., 2013; Lush and Piotrowski, 2014; Romero-Carvajal et l., 2015; Kozlovskaja-Gumbrienė et al., 2017; Dalle Nogare and Chitnis, 2017).”

- Figure 3K-O: By eye, dorsal mantle cell recovery doesn't look very robust. Is this a representative example? The total cell numbers in Figure 3Z don't really address if certain classes of cells fully recover.

The images shown throughout the paper are representative of the data collected from every sample. A variable number and distribution of mantle cells also occurs in unperturbed organs, and representative examples are shown in Figure 3K-O, as well as in Figure 3P-Y.

- Figure 4—figure supplement 1 and its related Discussion: I'm not quite sure what the authors claim regarding amount and timing of genetically encoded marker expression means. Are they using GFP reporter expression as a proxy for actual gene expression? Is so, it needs to be supported by endogenous expression data (e.g. insitus) or references to previous studies that demonstrate exogenous reporter expression reflects endogenous gene expression in these reporter lines.

We thank the reviewer for pointing out this ambiguty. To clarify, we did not intend to relate the behavior of the GFP reporter lines to changes in expression of endogenous genes, other than that of the Gateway57A transgene. We simply highlight the correlation between the behavior of *Sox2*:GFP and Gateway57A (also GFP) as suggestive of sustentacular-cell reversion to a primordial status during regeneration. The reviewer, however, makes us realize that this suggestion may be interpreted as a statement that is supported by additional data, which is not the case. Therefore, we have rephrased the entire section. Specifically, we eliminated the data that was used to indicate that SqGw57A represents a live sensor of supporting-cell maturity.

- Results: When describing the combination of reporter lines used, it would be helpful to the reader to put the name of the line next to the cellular structures they label in the text.

We have used Figure 1 for this purpose. We now specify when preseting the transgenic tools the color variants of the fluorescent makers and the identity of the cell types that are marked. When presenting Figure 1, we spell-out that “Specifically, the Tg[alpl:mCherry] line expresses cytosolic mCherry in the mantle and interneuromast cells (Figure 1D). The Et(krt4:EGFP)sqgw57A (hereafter SqGw57A) expresses cytosolic GFP in sustentacular cells (Figure 1E). The Tg[-8.0cldnb:LY-EGFP] (Cldnb:lynGFP) express a plasma-membrane targeted EGFP in the entire neuromast epithelium (Figure 1F), and the Tg[*Sox2*-2a-sfGFP] (*Sox2*:GFP) expresses cytosolic GFP in all the supporting cells (Figure 1G). For hair cells, we use Et(krt4:EGFP)sqet4 (SqEt4) that expresses cytosolic GFP (Figure 1H), or the Tg(myo6b:actb1-EGFP) (Myo6b:actin-GFP) that labels filamentous actin (Figure 1I).”

- Figure 4U-W; X-Z: By eye, the polarity of phalloidin labeling looks mixed, and X-Z is further confounded by the presence of immature hair-cells. To better make the case for fully reformed radial symmetry 1) a control, non-ablated NM should be shown for comparison 2) the axis of symmetry should be indicated with a hashed line 3) relative orientation of HC per NM should be quantified.

We are not sure what the reviewer refers to when writing “mixed”. A neuromast always carries one half of the hair cells pointing in the opposite direction of the other half. This is stated in the Introduction of the paper using Figure 1C, and exemplified in Figure 1I. We have introduced changes in Figure 1I and Figure 4W and 4Z to address this comment, by including a double-head arrow to indicate the axis of epithelial planar polarity, and the dual orientation of the hair cells along this axis.

- The observation that a few regenerating hair cells recovered planar polarity could be explained by Emx2 expression in 1 of 2 sibling hair cells (Jiang, Kindt and Wu, 2017). It's important to discuss this potential mechanism either here or in the Discussion section even if the authors didn't test for Emx2 expression in this study.

The reviewer correctly points to a recent paper by Jiang and collaborators about the role of Emx2 in hair-cell orientation. Our manuscript does not address planar cell polarity in particular, and thus did not originally contain a reference or a discussion of the relevance of Emx2. Specifically, we believe that Emx2 instructs the hair cells to implement the planar polarity cues to decide in which direction to polarize, without affecting coherent local polarity. Although the role of Emx2 in planar polarization is certainly fascinating, discussing it at length is beyond the scope of our work. Nevertheless, we recognize that many readers would find it relevant. Thus, we now include a paragraph contextualizing Emx2 with our results. Specifically, we state that: “…recent studies have identified a transcription factor called Emx2 that regulates the orientation of hair cells in neuromasts of the zebrafish. Emx2 is expressed in one half of the hair cells of the neuromast (those oriented towards the tail) and absent in the other half (which are coherently oriented towards the head). Loss- and gain-of-function of Emx2 alter planar cell polarity in a predictable manner because loss of Emx2 renders neuromasts with every hair cells pointing towards the head of the animal, and misexpression orients hair cells towards its tail. Because the coherent local axis of polarity is not affected by these genetic perturbations, Emx2 may act in hair cells as a decoder of global polarity cues. This evidence, together with our results, suggests that during neuromast regeneration founder cells autonomously organize the variegated expression of Emx2 in the regrowing eithelium with consequent recovery of a coherent axis of planar polarity and with one half of the hair cells pointing opposite to the other half”.

- Subsection “Sustentacular and mantle cells have different regenerative potential”, the conclusion that Sox2(+) sustentacular cells are tripotent is supported by evidence from a previous study; Sox2 has been shown to both expand progenitor cells and initiate their differentiation in the cochlea (Kemfle et al., Sci Rep, March 2016).

The reviewer points to a paper that analyzed *Sox2* in the mouse, whose inner ear does not possess the regenerative capacity of the neuromast. Specifically, Kemfle et al., (together with Millimaki, Sweet, and Riley, Dev. Biol. (2010) 338(2): 262) analyzed the role of *Sox2* in hair-cell embryonic development, forced production in post-embryonic murine ears, and their natural regeneration in zebrafish. The main conclusion of Kemfle et al., is that *Sox2* expands the pool of hair-cell progenitors, being more relevant to the specification of sensory hair cells, rather than that of non-sensory cells. We would like to note that we have not attempted to address hair-cell regeneration in our study, but rather the repair of the entire neuromast. Therefore, a direct extrapolation of the cited work to our results is not straightforward. Based on expression patterns of the endogenous *Sox2* in neuromasts and the expression of the transgenic *Sox2* sensor, we claim that *Sox2* is unlikely to play a role in specifying a sub-set of cells that would serve as a pool of progenitors dedicated to regeneration of sensory or non-sensory cells. Moreover, we would like to cite the work of Millimaki, Sweet, and Riley, Dev. Biol. (2010) 338(2): 262, in which it was shown that forced expression of *Sox2* in zebrafish expanded the epithelial domain that generates hair cells in the ear, but not so in neuromasts. This is now discussed in our manuscript. We now state that: “We propose that progenitor behavior is a facultative status that every sustentacular cell can acquire or abandon during regeneration.”

- In the same section, "predecing". Typo.

We thank the reviewer for identifying this error. It has been corrected by rephrasing the sentence.

- "Assessed spatial distribution… 60 hrs of regeneration" which ablation protocol did they use here?

We have used the same ablation protocol through the paper, which is now specifically stated in the Materials and methods section.

- Figure 5E: Both supporting cells and hair cell are labeled with GFP. What criteria did the authors use to differentiate supporting cells from hair cells?

The figure legend makes reference to hair cells, but these cells are not labeled in the accompanying images, which may have generated some confusion. In fact, it is precisely the lack of cytosolic green fluorescence in hair cells that we have used to quantify their number and pinpoint their localization during the lineage-tracing experiments. We now express this more clearly in the caption of the Figure 5.

- Figure 6K-L: Cell cycle length appears strongly correlated to regeneration time (K), but not so strongly correlated to neuromast size overall. The legend and the text states in L that the cell cycle length doesn't correlate well until 24 cells, but that doesn't seem directly apparent from the graph or the regression line.

We thank the reviewer for this comment. We specifically checked if the data is better described by a single regression, or by two regression lines with a ‘change point’ in between. Our model comparison accounts for the fact that the latter is more complex. Still, this model (weak correlation until 24 cells, strong correlation afterwards) explains the data significantly better than the single regression line.

- Subsection “Sox2(+) sustentacular cells are equipotent and plastic”: There is no mention of Figure 7C in the Results, and it's unclear to me from the figure legend how this data was generated.

We thank the reviewer for having identified this oversight on our part. The previous Figure 7C was mistakenly included because it showed data that were not used to support the conclusion of the paper. We have thus eliminated it.

- Discussion section; "suggesting that Sox2 itself is not essential" is overreaching. The observation made from interneuromast cells may show that Sox2 is not sufficient to drive regeneration, but it does not demonstrate it's not essential.

We agree with the reviewer. As stated above, the new version of our manuscript now says “We propose that progenitor behavior is a facultative status that every *Sox2*(+) sustentacular cell can acquire or abandon during regeneration. The *Sox2* transcription factor marks many progenitor cells and often drives stem-cell dependent regenerative processes in a variety of animals (Millimaki et al., 2010; Neves et al., 2013; Reinhardt et al., 2015; Sweet et al., 2011). *Sox2* is expressed widely in the neuromast, suggesting that *Sox2*(+) sustentacular cells are tri-potent progenitors.”

- Discussion section, second paragraph: Not mentioning the Emx2 study here is a glaring omission.

We have expanded the Discussion and include the references to Emx2, and we now write “Interestingly, recent studies have identified a transcription factor called Emx2 that regulates the orientation of hair cells in neuromasts of the zebrafish. Emx2 is expressed in one half of the hair cells of the neuromast (those oriented towards the tail) and absent in the other half (which are coherently oriented towards the head). Loss- and gain-of-function of Emx2 alter planar cell polarity in a predictable manner because loss of Emx2 renders neuromasts with every hair cells pointing towards the head of the animal, and misexpression orients hair cells towards its head. Because the coherent local axis of polarity is not affected by these genetic perturbations, Emx2 may act in hair cells as a decoder of global polarity cues. This evidence, together with our results, suggests that during neuromast regeneration founder cells autonomously organize the variegated expression of Emx2 in the regrowing eithelium with consequent recovery of a coherent axis of planar polarity and with one half of the hair cells pointing opposite to the other half. The future development of live markers of Emx2 expression will be able to test this prediction.”

- The Discussion overall would benefit from a more thorough incorporation of molecular mechanisms reported in previous studies examining supporting cell proliferation, differentiation, and repolarization in hair-cell organs to understand how this study's results enriches our understanding of the self-regulatory processes that guide lateral-line organ restoration.

Following the reviewer’s advice, we have substantially revised the Discussion and now include a sentence, fully referenced, about intercellular signaling pathways involved in neuromast formation and the regeneration of hair cells.

Reviewer #2:This study focus on one of the most intriguing and fascinating aspects or regenerative biology: the accuracy with which organs respond to injuries to restore proper size and cellular organization. The authors use neuromasts of fish to explore this phenomenon and fully exploit the advantages of the system. The videos and figures are of very high quality and aesthetic value, and the authors extract from them an exhaustive quantification of cellular responses during regeneration. I find the manuscript nicely written and in most sections easy to follow.I failed, however, to understand what is the main new finding that the authors are presenting. It is known from several models that tissues (epidermis), organs (liver, skin, the fish heart), limbs (axolotls and newts, caudal fin in fish), systems (hematopoietic) and even organisms (planarians) can re-generate accurately to their pre-injury state. The authors state clearly some of the open questions in the field, but it feels they fall short at addressing them thoroughly when using the neuromast. The major achievement of the manuscript is the acquisition of an extensive dataset on individual cellular behavior during the regeneration response. The two main points I see are that neuromasts regenerate what is lost, and cell identity is acquired along a radial axis. Unfortunately and despite the extensive analysis performed and the quality of the data, it is hard to understand the conceptual novelty of this study.

We share the reviewer’s enthusiasm about the biological problem that we investigate and the significance of our results, and are grateful for the encouraging words.

Major comments:During the Introduction and the initial section of Results, it feels that the authors neglect several findings from a previous publication of another group, Sánchez et al., 2016, on the regeneration of neuromasts upon ablation of the entire organ. The results presented in the submitted paper do not match a proportion of the results previously published (neuromast regeneration upon ablation or pharmacological inactivation of Schwann cells). The authors should explain this discrepancy and acknowledge that organ regeneration after severe injuries can indeed happen from other sources as well.

We have cited Sánchez et al., 2016 in the Results section of our initial submittal. Because we could not replicate the findings of Sánchez et al., our data led us to a different conclusion. One possibility that may explain this discrepancy is the differences in the cell-ablation methods and/or the reporters used to identify Schwann cells. Nevertheless, the new focus of our work has made this part of the original paper no longer necessary, and was therefore suppressed. Please, note that this change does not alter any of the main conclusions of our work.

The authors state that inter neuromast cells are not necessary for the regeneration response. They show however that in their absence, organs regenerate to a smaller size. What is then the role of interneuromast cells?

We state that the interneuromast cells are not necessary for regeneration, but did not try to imply that they play no role in the process. We are more specific in the revised version of our manuscript, where we now write: “However, *Sox2* is expressed widely in the neuromast, including the interneuromast cells that are not essential to neuromast repair. This observation suggests that *Sox2* itself may not be essential for regeneration.” Additionally, we now indicate, on pp8., that “Interneuromast cells are not essential for neuromast regeneration in larval zebrafish, although they may contribute to mantle cell re-emergence”, and on pp9 that “These observations reinforce our previous suggestion that interneuromast cells have a non-essential but appreciable contribution to regeneration.”

In reference to the autonomous regeneration of epithelial polarity the authors show that neuromasts regenerate the proper polarisation after severe injuries in the presence of interneuromast cells, or after ablation of hair cells in the absence of interneuromast cells. To claim an autonomous process, they should look at the polarisation plane after severe injuries and in the absence of interneuromast cells – experiments done for a previous section but not analyzed for epithelial polarity.

We agree with the reviewer, and we no longer make this claim.

In addition, the authors should consider that anisotropic forces might originate not only from interneuromast cells and therefore evaluate whether this is the valid conclusion from their experimental data.

The reviewer is correct in pointing that we have neglected potential additional sources of mechanical forces. In the case of planar polarity in particular, forces by the interneuromast cells are obvious candidates because of the position of these cells relative to the polarity axis. Yet, it is true that other forces may play an instructive role, and this is something that we have not analyzed because neither our own work, nor the literature, suggests candidates that we can test experimentally. Therefore, we now state “To test if plane-polarizing cues derive from anisotropic forces exerted by the interneuromast cells that are always aligned to the axis of planar polarity of the neuromast epithelium, we ablated these cells flanking an identified neuromast, and concurrently killed the hair cells with the antibiotic neomycin (Figure 4X-Y). In the absence of interneuromast cells regenerating hair cells recovered normal coherent planar polarity (n=16), suggesting the existence of alternative sources of polarizing cues (Figure 4Z). Collectively, these findings reveal that as few as 4 supporting cells can initiate and sustain integral organ regeneration.”

The authors emphasize the accuracy of the regenerative response, but this seems to be restricted to the distribution of cell types. The number of cells, and therefore the size of the organ, is restored only to a 70% after 7dpi. This needs to be stressed, taken in consideration for the conclusions, and contemplated in the discussion.

The reviewer recognizes that the focus of our work has been directed towards the recovery of cell classes, their spatial distribution, their relative numbers and of hair-cell polarity, and that we have not focused on organ size. The reason behind this emphasis is that the recovery of organ proportions and geometry remain far lesser understood biological problems. We show that cell-fate acquisition and cell-class distribution are not tissue-size dependent. We also reflect upon our data and clearly indicate that: “Interneuromast cells are not essential for neuromast regeneration in larval zebrafish, although they may contribute to mantle cell re-emergence”.

Additionally, the new version of the manuscript highlights and strengthen this focus, but also briefly touches-upon organ size in the discussion, as stated above: “Moreover, we did not observe regenerative overshoot of any cell class (Agarwala et al., 2015), suggesting the existence of a mechanism that senses the total number of cells and the cell-class balance during tissue repair (Simon et al., 2009). Previous work indicates that such mechanisms may be based on interplay between FGF, Notch and Wnt signaling (Ma et al., 2008; Wibowo et al., 2011; Wada et al., 2013; Romero-Carvajal et l., 2015; Dalle Nogare and Chitnis, 2017).”

It is still not clear whether neuromast cells acquire a proper position during regeneration or whether they acquire their fate based on the position they occupy. The authors state both claims along the manuscript although they are mutually exclusive.

We thank the reviewer for this comment because it is central to the problem of regeneration of a complex structure in vivo. We understand that the reviewer finds some ambiguity in our discussion, specifically whether cells acquire their fate based on position, or position based on fate. Put in other words, whether there is a hierarchical relationship between these two features and, if so, which is their relationship. Our careful tracing of cell fate and machine-learning based analyses of cellular behavior are meant to specifically address this important point. By evaluating patterns of cellular behavior, we find no evidence of directional cell movement or intercalation to support the notion that cells acquire position based on a pre-defined fate. We also found no evidence of incorrectly-localized cell classes being extruded from the epithelium or eliminated by apoptosis. Importantly, by tracking clone-growth trajectories we find that the cells acquire fate based on their localization in the epithelium and, specifically, their position along the mediolateral axis of the epithelium.

The claim that sustentacular cells revert to an embryonic (undifferentiated) state immediately after tissue injury is based in the transient down-regulation of a single fluorescent reporter. The authors should present data for some of the many additional markers or tone down their conclusions. On a more general view, I disagree with the claim that undifferentiated equals embryonic.

Reviewers 1 and 2 have an identical concern about this point. We did not intend to relate the behavior of the GFP reporter lines to changes in endogenous gene expression, other than the transgenes themselves. Our highlighting the correlation between the expression of *Sox2*:GFP and Gateway57A (also GFP) during early development of the lateral line was done to indicate a possible reversion of sustentacular cells to a primordial status during regeneration. The current version of the manuscript, with its new focus, no longer makes this claim.

Reviewer #3:The work by Oriol Viader-Llargués et al. examines how zebrafish mechanosensory organs, neuromasts (NMs), respond to injury. The authors ablate portions of various sizes from different regions of the organ and then examine organ regeneration using live imaging of transgenes that mark distinct cell lineages within NMs. A number of previous studies examined regeneration of a single NM cell type, hair cells; however, regeneration capacity of other cell types has not been studied. While carefully done, the study is largely descriptive and does not build on the previous knowledge of molecular mechanisms that drive NM development and regeneration. Nevertheless, this work presents a number of novel findings, related to the cellular mechanisms of organ regeneration that should contribute significantly to the field of organ regeneration: 1) NMs can fully regenerate up 70 to 90% of their original size regardless of injury site; 2) regeneration process restores relative proportion and proper polarity of NM cell types; 3) regeneration is based on a resident population of Sox2-positive cells (not on stem cells) that restore all cell types within NMs; 4) even severely damaged organs (as few as 4 cells left) can fully regenerate; 5) cell fate during regeneration can be predicted based on the distance of dividing progenitors from the center. I have a few specific comments related to data interpretation and conclusions.

We are happy to learn that the reviewer finds that our work presents a number of novel findings related to the cellular mechanisms of organ regeneration, and that it will contribute significantly to the field of organ regeneration.

As the authors point out, previous studies looking at hair cell regeneration found that the source of hair cell progenitors resides in NM poles. Present study found that Sox2^+^ progenitors throughout the NM contribute to hair cells. How do they reconcile the differences? This should be discussed.

We do not find any contradiction between our findings and those that have led to the conclusion that there are regional differences in supporting-cell behavior, specifically because early studies have focused on hair-cell regeneration rather than whole-neuromast repair. Our finding that sustentacular cells are tripotent progenitors and that the neuromast epithelium is symmetric in its regenerative ability do not contradict previous conclusions.

Ablation of inter-NM cells led to smaller regenerating NM after injury. Again, this issue is not discussed.

This is indeed possible. Thus, we now state that: “Interneuromast cells are not essential for neuromast regeneration in larval zebrafish, although they may contribute to mantle cell re-emergence”.

The authors conclude that Sox2^+^ cells are resident progenitors that potentially dedifferentiate following injury and then contribute to all three distinct cells types in the NM. Have they tried eliminating Sox2^+^ cells to support this assertion?

This reviewer shares a concern with Reviewer 2 about the role of *Sox2* cells during neuromast regeneration. Every cell, other than the hair cells, express *Sox2*. The hair-cells are post-mitotic and likely need supporting cells to remain in the epithelium. Therefore, eliminating all *Sox2*-positive cells will result in the elimination of the entire neuromast, which we have done and shown to lead to permanent organ loss. Instead, we support our assertion by tracking the fate of *Sox2*(+) cells using clonal analysis, live imaging and machine learning, which reveal that *Sox2*(+) supporting cells, specifically sustentacular cells, proliferate and differentiate in all three cells classes during neuromast repair.

Related to the last point, it would certainly add to the impact of the paper if the authors can show that Sox2^+^ cells de-differentiate, based on molecular markers.

We agree with the reviewer, but this is a difficult problem to address. The reason is that it is not clear what markers can be used to examine supporting-cell de-differentiation. In many sensory organs that include the ear and the lateral line, the transcription factor Atoh1 is a marker of committed but not terminally differentiated pro-sensory cells. We have reported previously that in neuromasts Atoh1 is only transiently expressed by supporting cells under low Notch signaling, and that it eventually becomes stabilized in hair-cell progenitors. During neuromast embryonic development, the chemokine receptor CXCR4b is expressed in the lateral line primordium, specifically at its front where uncommitted neuromast progenitor cells are located. Its expression ceases once neuromasts mature despite the continuous production of cells. Thus, CXCR4b cannot be assumed as a de-differentiation marker. Therefore, because of this lack of appropriate markers, the new version of the manuscript no longer makes the original claim about the de-differentiation of *Sox2*(+) cells during regeneration.

[Editors’ note: the author responses to the re-review follow.]

Reviewer #1:The revised version of the manuscript by Oriol Viader-Llargués et al. is significantly improved from the previous version. Its message is more focused, its conclusions better supported by the data, and references to relevant work were incorporated.With that said, I have a few major questions that don't necessarily need to be addressed with additional experiments:1) The mantle cell linage data show that the mantle cells do not give rise to any cell class other than their own, leading to the conclusion that mantle cells are not essential contributors to neuromast repair. Yet none of their injury paradigms leave only the mantle cells intact i.e. every injury with mantle cells present also has sustentacular cells present. It could be the case that mantle cells are essential to regenerating a neuromast if the sustentacular cells are absent.

The reviewer is correct in arguing that we have not formally ruled out a contribution of mantle cells to neuromast regeneration, specifically when every other cell class is gone. Although such experiment will definitely be revealing, we have been unable to do it in a controlled manner for technical reasons. In light of this, we now include a slightly modified statement about the behavior of the mantle cells under our experiments, as well as their possible role under the condition identified by the reviewer. We now state in the Discussion:

“The behavior of the mantle cells is especially intriguing. Complete elimination of parts of the lateral line by tail-fin amputation have revealed that mantle cells are able to proliferate and generate a new primordium that migrates into the regenerated fin to produce new neuromasts (Dufourcq et al., 2006). This observation can be interpreted as suggesting that under some injury conditions, mantle cells are capable of producing all the cell classes of a neuromast. Transcriptomic profiling of mantle cells following neuromast injury revealed that these cells up-regulate the expression of multiple genes (Steiner et al., 2014). Furthermore, a recent study has revealed that mantle cells constitute a quiescent pool of cells that re-enters cell cycle only in response to severe depletion of sustentacular cells (Romero-Carvajal et al., 2015), suggesting that these cells may conform a stem-cell niche for proliferation of sustentacular cells. Thus, the collective evidence indicates that the mantle cells respond to damage and contribute to the regenerative processes, and may drive the regeneration of an entire organ if every other cell class is lost.”

The authors refer to a study in their rebuttal that suggests mantle cells are capable of producing all neuromast cell classes in a regenerating fin. Do the authors predict this would be the case, if they ablated everything but the mantle cells? If not, why?

As stated above, we believe that this may well be the case, and have added the above paragraph in the Discussion to specifically address this point. Because this possibility is largely based on previous work, we have included the relevant references: Dufourcq et al., 2006; Steiner et al., 2014; Romero-Carvajal et al., 2015.

2) The authors discussed the potential influence of interneuromast cells on planar polarity, but did not elaborate influence of INCs on neuromast size (Figure 4K-P). The result is striking and should be discussed further than the statement in the Results-"non-essential, yet appreciable contribution to regeneration".

Please, note that this experiment equates size with cell count. We have added language to clarify this point and address the reviewer’s comment:

“We find that interneuromast cells are not essential for neuromast regeneration because severely damaged organs recover all cell classes in the appropriate localization in the absence of interneuromast cells. However, we systematically observed smaller organs when interneuromast cells where ablated. These observations suggest that these peripheral cells may yet help regeneration, either directly by contributing progeny, or by producing mitogenic signals to neuromast-resident cells”.

Reviewer #2:In the new version of this manuscript, Viader-Llargues, Lopez-Schier, and colleagues present their results with a more clear aim, focusing on quantitative aspects of the regeneration response and new tools to investigate it. My main points were properly addressed, namely toning down certain conclusions, including alternative scenarios and focusing on defined topics rather than superficially following many. I appreciate particularly the scientific quality and dedication of the authors' replies to many of my previous concerns. The lack of overshooting of cell types during the regeneration response was a very nice addition that contrasts to other regenerative systems, and illustrates how controlled the regenerative response is in neuromasts.These are my comments on the current version:In the title and along the manuscript the authors use the term "cell-phylogeny tracing". Is there any reason not to use the widely accepted term "lineage" instead? I feel that "lineage" works better in this context since it reflects more accurately the continuum of the data that the authors have acquired.

Under a standard definition of lineage and phylogeny, it is true that what we have traced is cellular lineages: “A lineage is a series of cells that can be connected directly by a continuous line of descent from a primordial progenitor”.

We agree with the reviewer that lineage is an optimal word given its widely accepted meaning in the field. Yet, we chose the word phylogeny because we have also shown the relationship between the lineages of several initial primordial progenitors (founder cells). Having said this, however, we understand that “phylogeny” may generate confusion among the readership and accept the critique. Thus, we have replaced phylogeny with lineage throughout the text, including the title.

The issue of self-regulation of the regenerative response is dealt more carefully than in the previous version. Still, I would like the authors to be more explicit about their interpretation of the data, mainly on the Discussion. One of the aspects the authors focus on is the re-establishment of a polarity axis, and this is a very interesting aspect of the regeneration response since polarity is heterogeneous among different neuromasts of the posterior lateral line – unlike cell types and their distribution. In other words: neuromasts have one solution for the cell-type problem, but two solutions for the polarity problem. Previous work from the group (Lopez-Schier et al., 2004 and Lopez-Schier and Hudspeth, 2006) has clearly stated that the polarity axis is related to the final migration of the neuromasts during organogenesis. I find troubles imagining how a self-renewing organ (or 4-10 cells, to put it bluntly) will always choose the original polarity in the absence of external cues. I agree with the authors that external cues have minor roles in the establishment of cells types or position of cells, but extending self-organization to the re-establishment of polarity seems inappropriate to me. Am I missing something? Maybe the authors could speculate about the role of the afferent axons (I guess they should remain to some extent under the injury paradigms used in this study), which they have shown to display a high accuracy recognizing hair cells of a given polarity (Pujol-Marti, Current Biology 2014).

The reviewer has identified one of the most mysterious and, in our opinion, fascinating problem of architectural recovery during organ regeneration. The collective evidence from previous work of our group and others, and evidence that we present here indicates that planar polarity has two components: the local coherent orientation of hair cells along a single axis (in horizontal neuromasts either rostrally or caudally), and the global orientation of this axis relative the main body axes of the fish. The local coherent orientation involves the core planar polarity pathway (López-Schier et al., 2004), as well as a segregated activity of the Emx2 transcription factor (Jiang et al., 2017). We show that the local coherent orientation of hair cells is self-regulatory. We do not know for certain what controls the global orientation of the hair cells, other than that it is initially determined by the direction of movement of the lateral-line primordium (López-Schier et al., 2004). We have always stated that architectural repair (including planar polarity) is achieved with “minimal” extrinsic information (Abstract). We do argue that self-organization is an optimal morphogenetic process to govern organ repair (Discussion), but discuss where and when self-organization may occur, make a clear distinction between self-organization and self-assembly, and do not indicate that a purely self-organizing process is at play. Therefore, we now clarify that when referring to planar polarity we exclusively focus on the local coherent polarization of the hair cells, and further state that:

“We would like to highlight that we do not currently understand the global polarization of the neuromast epithelium relative to the main body axes of the animal. External sources of polarity may impinge in the recovery of these global axes during neuromast regeneration. Previous work has demonstrated that local and global polarization occur independently of innervation López-Schier and Hudspeth, 2006), but other potential polarizing cues remain untested. Therefore, at present we can only support the notion that local coherent polarity is self-organizing, whereas global orientation may be controlled externally.”

I particularly enjoyed the reply of the authors regarding the differences and similarities in mantle cell behavior during homeostasis and regeneration. I think it would be an added value if they incorporate these concepts in the Discussion.

We are glad to read this. Reviewers 1 and 2 have expressed a similar feeling. Thus, as stated above, we now write in the Discussion:

“The behavior of the mantle cells is especially intriguing. Complete elimination of parts of the lateral line by tail-fin amputation have revealed that mantle cells are able to proliferate and generate a new primordium that migrates into the regenerated fin to produce new neuromasts (Dufourcq et al., 2006). This observation can be interpreted as suggesting that under some injury conditions, mantle cells are capable of producing all the cell classes of a neuromast. Transcriptomic profiling of mantle cells following neuromast injury revealed that these cells up-regulate the expression of multiple genes (Steiner et al., 2014). Furthermore, a recent study has revealed that mantle cells constitute a quiescent pool of cells that re-enters cell cycle only in response to severe depletion of sustentacular cells (Romero-Carvajal et al., 2015), suggesting that these cells may conform a stem-cell niche for proliferation of sustentacular cells. Thus, the collective evidence indicates that the mantle cells respond to damage and contribute to the regenerative processes, and may drive the regeneration of an entire organ if every other cell class is lost.”

Results section, paragraph two; I believe that the authors should include a reference to Grant et al., 2005. In fact, the authors do so in their response to Reviewer #1.

This is correct. We have added the reference.

Subsection “Sustentacular cells are equipotent and plastic”. I found this part difficult to understand. Either the written numbers do not match what the authors show between brackets, or there is some mistake in the annotation. What is "peak at 8h, mean+-s.d at 14+-9 hours"? Overall, I find this part the less clearly written of the manuscript. Also, is it valid to call a peak sharp with such a big s.d.?

We see how the previous expression of this result may be difficult to understand. We now state it more clearly as:

“Cell-cycle length in the 1^st^ generation peaks around 10 hours (9.8 ± 3.3h, median ± interquartile range (iqr)) (Figure 7C), but it begins to increase and to vary in the 2^nd^ generation (11.5 ± 7.3h, median ± iqr), and more so in the 3^rd^ generation (18.8 ± 20.3h, median ± iqr)”.

Discussion paragraph one: The authors state that sustentacular cells are equipotent, which I think is an overstating.The data presented in the manuscript reveals that 4 or more of them are enough to regenerate the entire organ. The use of equipotency, in my view, states that all of them do the same during the process. In their accurate data in Figure 7, they show that some of them generate M cells, some other S cells, and some other combinations of H and other cell types. I understand that equipotent members of a population could behave in different manners based on either stochastic internal programs or external cues, but to prove which is the case demands a more extensive dataset and a deep mathematic analysis, which is not the scope of the present study – although this is a fantastic system to tackle it! I feel that the use of equipotent in this context assumes features that were not tested experimentally. I suggest the authors stay with "tri-potent", a term that they have used along Results.

This is correct. We have deleted the term equipotent and use now tri-potent.

Reviewer #3:This is a revised version of a manuscript by Oriol Viader-Llargués et al. that deals with cellular mechanisms of neuromast regeneration. The previous version was criticized by its descriptive nature, lack of focus and omission of some references. The authors largely remedied these issues, although I still feel that some of their findings are not sufficiently discussed in the context of previous studies.Comments:The authors mention that their findings contrast those of Sanchez et al. study, but do now offer any discussion as to why that is the case. I think it is important to offer at least some explanation.

We believe that these discrepancies may be a result of differences in the experimental approaches and/or the markers used to visualize cells. Another possible explanation is the age of the animals that were used in either study. Sánchez et al., used 3dpf larvae, whereas all our ablations were performed in older animals. These differences, alone or combined, may account for the differences in the outcome or interpretation of the two studies. Recognizing this, on pp.7 we do mention that differences in age may explain the differences between both studies. However, we now state more explicitly in the Discussion:

“We show that the complete elimination of a neuromast is irreversible in larval zebrafish. However, Sánchez and colleagues have previously reported that interneuromast cells can generate new neuromasts (Sánchez, 2016). By assaying DNA synthesis prior to mitosis, we show that interneuromast cells do not proliferate after neuromast ablation. These differences may be explained by differences in ablation protocols (electroablation versus laser-mediated cell killing), the age of the specimens (embryos versus early larva) or the markers used to assess cellular elimination.”

It seem that examples of severe ablation following full regeneration always leave at least one mantle cell (“Neuromast architecture recovers after severe loss of tissue integrity”). If this is indeed the case, it may indicate that a combinatorial signal(s) is required to initiate organ regeneration.

This is correct. Yet, by analyzing neuromasts completely devoid of mantle cells (Figure 3), we show that neuromast regenerate to a normal status. Therefore, we do not think that combinatorial signaling between mantle and sustentacular cells is essential for proper repair.

There are only two cases when sustentacular cell progenitors gave rise to all three cell types. Based on these small numbers, I think the authors need to be careful concluding that a sustentacular cell is a multipotent progenitor for all cell types in the neuromast.

This is correct. Our current and previous work show that sustentacular cells can re-generate and also generate hair cells. We show that they can also give rise to mantle cells. Thus, the sustentacular-cell population is tri-potent. The question is whether each individual sustentacular cell is also tri-potent. Our clonal analysis reveals that this is the case, and makes us believe that the fact that some events are rare does not invalidate the tri-potentiality idea. We have been careful to restrict our conclusions to the available data and trust that we had not overstated our claims. Nevertheless, we have now changed a subtitle to “The sustentacular-cell population is tri-potent and plastic”.

The finding that at least 4 cells are needed to reconstitute an organ is intriguing. This is reminiscent of planarian organ regeneration where 1/300th part but not less can reconstitute full animal. It was later discovered that this is a minimum fraction of the animal roughly containing at least one stem cell (neoblast) necessary to regenerate all tissue lineages. Again, some discussion, as to why the authors think this is case (4 cells but not fewer are required) is warranted

This is indeed very interesting. We have found technically challenging to leave less than 4 cells in a consistent manner without eliminating the entire neuromast. Therefore, we cannot rule out the possibility that as little as a single founder cells may be able to regenerate an entire organ. We have included a sentence in the Discussion about this point

“It is technically challenging to consistently maintain fewer than 4 cells in toto without eliminating the entire neuromast. Thus, we cannot rule out the possibility that a single founder cell may be able to regenerate a neuromast”.

Discussion paragraph two, final sentence: Seleit et al., 2017 showed that "new cell type" exists in zebrafish. They also showed that mantle cell are neuromasts stem cells. Thus, it is important to discuss how homeostatic cell renewal differs from cell renewal during NM regeneration (i.e. mantle cell as stem cell during homeostasis vs. sustentacular cell as a multipotent progenitor during regeneration). This is an interesting question, as there are examples where cells can be driven to change their fate by extreme injury paradigm.

This is indeed very interesting. Seleit et al., 2017 have revealed a "new cell type" in Medaka and zebrafish neuromasts. As stated for reviewers 1 and 2, we have now added a section in the Discussion with a thorough explanation of mantle cell behaviors reported in the literature and how they compare with our findings.